# The Malina oceanographic expedition: How do changes in ice cover, permafrost and UV radiation impact biodiversity and biogeochemical fluxes in the Arctic Ocean?

Philippe Massicotte[1], Rainer M.W. Amon[2,3], David Antoine[4,5], Philippe Archambault[6], Sergio Balzano[7,8,9], Simon Bélanger[10], Ronald Benner[11,12], Dominique Boeuf[7], Annick Bricaud[5], Flavienne Bruyant[1], Gwenaëlle Chaillou[13], Malik Chami[14], Bruno Charrière[15], Jingan Chen[16], Hervé Claustre[5], Pierre Coupel[1], Nicole Delsaut[15], David Doxaran[5], Jens Ehn[17], Cédric Fichot[18], Marie-Hélène Forget[1], Pingqing Fu[19], Jonathan Gagnon[1], Nicole Garcia[20], Beat Gasser[21], Jean-François Ghiglione[22], Gaby Gorsky[5], Michel Gosselin[13], Priscillia Gourvil[23], Yves Gratton[24], Pascal Guillot[13], Hermann J. Heipieper[25], Serge Heussner[15], Stanford B. Hooker[26], Yannick Huot[27], Christian Jeanthon[7], Wade Jeffrey[28], Fabien Joux[22], Kimitaka Kawamura[29], Bruno Lansard[30], Edouard Leymarie[5], Heike Link[31], Connie Lovejoy[1], Claudie Marec[1,32], Dominique Marie[7], Johannie Martin[33], Jacobo Martín[34,35], Guillaume Massé[1,36], Atsushi Matsuoka[1], Vanessa McKague[37], Alexandre Mignot[5,38], William L. Miller[39], Juan-Carlos Miquel[21], Alfonso Mucci[40], Kaori Ono[41], Eva Ortega-Retuerta[22], Christos Panagiotopoulos[20], Tim Papakyriakou[17], Marc Picheral[5], Dieter Piepenburg[42,43], Louis Prieur[5], Patrick Raimbault[20], Joséphine Ras[5], Rick A. Reynolds[44], André Rochon[13], Jean-François Rontani[20], Catherine Schmechtig[45], Sabine Schmidt[46], Richard Sempéré[20], Yuan Shen[11,47], Guisheng Song[48,49], Dariusz Stramski[44], Dave Stroud G.[50], Eri Tachibana[41], Alexandre Thirouard[5], Imma Tolosa[21], Jean-Éric Tremblay[1], Mickael Vaïtilingom[51], Daniel Vaulot[7,52], Frédéric Vaultier[20], John K. Volkman[53], Huixiang Xie[13], Guangming Zheng[44,54,55], and Marcel Babin[1]

[1]Takuvik Joint International Laboratory / UMI 3376, ULAVAL (Canada) - CNRS (France), Université Laval, Québec, QC, Canada
[2]Department of Marine and Coastal Environmental Science, Texas A&M University Galveston Campus, Galveston, Texas, 77553, USA
[3]Department of Oceanography, Texas A&M University, College Station, Texas, 77843, USA
[4]Remote Sensing and Satellite Research Group, School of Earth and Planetary Sciences, Curtin University, Perth, WA 6845, Australia
[5]Sorbonne Université, CNRS, Laboratoire d'Océanographie de Villefranche (LOV) / UMR 7093, F-06230 Villefranche-sur-Mer, France
[6]ArcticNet, Québec-Océan, Takuvik Joint International Laboratory / UMI 3376, ULAVAL (Canada) - CNRS (France), Université Laval, Québec, QC, Canada
[7]Sorbonne Université, CNRS, Station Biologique de Roscoff - Adaptation et Diversité en Milieu Marin / UMR 7144, 29680 Roscoff, France
[8]Present address: Stazione Zoologica Anton Dohrn Napoli (SZN), Naples, Italy
[9]NIOZ Royal Netherlands Institute for Sea Research, Den Burg, Netherlands
[10]Département de Biologie, Chimie et Géographie (groupes BORÉAS et Québec-Océan), Université du Québec à Rimouski, Rimouski, QC, Canada
[11]School of the Earth, Ocean and Environment, University of South Carolina, Columbia, South Carolina, 29208, USA

[12]Department of Biological Sciences, University of South Carolina, Columbia, South Carolina, 29208, USA

[13]Québec-Océan, Institut des sciences de la mer de Rimouski (ISMER), Université du Québec à Rimouski, Rimouski, QC, Canada

[14]Sorbonne Université, CNRS, Laboratoire Atmosphères Milieux Observations Spatiales (LATMOS) / UMR 8190, Boulevard de l'Observatoire, CS 34229, 06304 Nice Cedex, France

[15]Université de Perpignan Via Domitia (UPVD), CNRS, Centre de Formation et de Recherche sur les Environnements Méditerranéens (CEFREM) / UMR 5110, 52 Avenue Paul Alduy, 66860 Perpignan Cedex, France

[16]SKLEG, Institute of Geochemistry, Chinese Academy of Sciences, 99 West Lincheng Road, Guiyang, Guizhou 550081, P.R. China

[17]Centre for Earth Observation Science, Department of Environment and Geography, University of Manitoba, Winnipeg, MB, Canada

[18]Department of Earth and Environment, Boston University, Boston, Massachusetts, 02215, USA

[19]Institute of Surface-Earth System Science, Tianjin University, Tianjin, China

[20]Aix Marseille Université, Université de Toulon, CNRS, IRD, MIO UM 110, 13288 Marseille, France

[21]International Atomic Energy Agency (IAEA) / Environment Laboratories, MC98000, Monaco, Monaco

[22]Sorbonne Université, CNRS, Laboratoire d'Océanographie Microbienne (LOMIC) / UMR 7621, Observatoire Océanologique de Banyuls, France

[23]Sorbonne Université, CNRS, Station Biologique de Roscoff - Centre de recherche et d'enseignement en biologie et écologie marines / FR2424, 29680 Roscoff, France

[24]Institut national de la recherche scientifique - Centre Eau Terre Environnement (INRS-ETE), Québec, QC, Canada

[25]Department of Environmental Biotechnology, Helmholtz Centre for Environmental Research - UFZ, Permoserstraße 15, D-04318 Leipzig, Germany

[26]Ocean Ecology Laboratory, NASA Goddard Space Flight Center, Greenbelt, MD, United States

[27]Département de géomatique appliquée, Université de Sherbrooke, Sherbrooke, QC, Canada

[28]Center for Environmental Diagnostics & Bioremediation, Universty of West Florida, 11000 University Parkway, Pensacola, FL 32514 USA

[29]Chubu Institute for Advanced Studies, Chubu University, Kasugai, Japan

[30]IPSL and Université Paris-Saclay, CEA-CNRS-UVSQ, Laboratoire des Sciences du Climat et de l'Environnement (LSCE) / UMR 8212, 91190 Gif-sur-Yvette, France

[31]Department Maritime Systems, University of Rostock, 18059 Rostock, Germany

[32]Université de Bretagne Occidentale - UBO, CNRS, IRD, Institut Universitaire Européen de la Mer (IUEM) / UMS 3113, 29280 Plouzané, France

[33]Québec-Océan & Département de biologie, Université Laval, Québec, QC, Canada

[34]Centro Austral de Investigaciones Científicas (CADIC-CONICET), Houssay 200, 9410 Ushuaia, Argentina

[35]ICPA-UNTDF, Ushuaia, Argentina

[36]Station Marine de Concarneau, MNHN-CNRS-UPMC-IRD, Laboratoire d'océanographie et du climat : expérimentations et approches numériques (LOCEAN) / UMR 7159, 29900 Concarneau, France

[37]Center for Marine and Environmental Studies, University of the Virgin Islands, St. Thomas, VI, USA

[38]Mercator Ocean International, Parc Technologique du Canal, 8-10 rue Hermès – Bâtiment C, 31520 Ramonville Saint-Agne, France

[39]Department of Marine Sciences, University of Georgia, 325 Sanford Dive, Athens, GA 30602

[40]GEOTOP and Department of Earth and Planetary Sciences, McGill University, Montréal, QC, Canada

[41]Institute of Low Temperature Science, Hokkaido University, Sapporo, 060-0819, Japan

[42]Alfred Wegener Institute, Helmholtz Centre for Polar and Marine Research, Am Handelshafen 12, 27570 Bremerhaven, Germany

[43]Helmholtz Institute for Functional Marine Biodiversity at the University of Oldenburg, Ammerländer Heerstraße 231, 26129 Oldenburg, Germany

[44]Marine Physical Laboratory, Scripps Institution of Oceanography, University of California San Diego, La Jolla, CA, 92093-0238 USA

[45]Sorbonne Université, CNRS, Ecce Terra Observatoire des Sciences de l'Univers (OSU) - UMS 3455, 4, Place Jussieu 75252 Paris Cedex 05, France
[46]Université de Bordeaux, CNRS, OASU, Environnements et Paléoenvironnements Océaniques et Continentaux (EPOC) / UMR 5805, F-33615 Pessac, France
[47]State Key Laboratory of Marine Environmental Science, College of Ocean and Earth Sciences, Xiamen University, Xiamen, Fujian, P. R. China
[48]Institut des sciences de la mer de Rimouski (ISMER), Université du Québec à Rimouski, Rimouski, QC, Canada
[49]School of Marine Science and Technology, Tianjin University, Tianjin, 300072, China
[50]Department of Physics, Ohio State University, Columbus, Ohio 43210, USA
[51]Université des Antilles Pointe-à-Pitre, Laboratoire de Recherche en Géosciences et Energies (LARGE) / EA 4539, Guadeloupe, France
[52]Asian School of the Environment, Nanyang Technological University, Singapore
[53]CSIRO Marine and Atmospheric Research and CSIRO Wealth from Oceans Nationa lResearch Flagship, GPO Box 1538, Hobart, Tasmania 7001, Australia
[54]NOAA/NESDIS Center for Satellite Applications and Research, 5830 University Research Court, College Park, MD 20740, USA
[55]Earth System Science Interdisciplinary Center, University of Maryland Research Park, 5825 University Research Court, College Park, MD 20740, USA

**Correspondence:** Marcel Babin (marcel.babin@takuvik.ulaval.ca)

**Abstract.** The MALINA oceanographic campaign was conducted during summer 2009 to investigate the carbon stocks and the processes controlling the carbon fluxes in the Mackenzie River estuary and the Beaufort Sea. During the campaign, an extensive suite of physical, chemical and biological variables was measured across seven shelf–basin transects (south-north) to capture the meridional gradient between the estuary and the open ocean.

Key variables such as temperature, absolute salinity, radiance, irradiance, nutrient concentrations, chlorophyll *a* concentration, bacteria, phytoplankton and zooplankton abundance and taxonomy, and carbon stocks and fluxes were routinely measured onboard the Canadian research icebreaker *CCGS Amundsen* and from a barge in shallow coastal areas or for sampling within broken ice fields. Here, we present the results of a joint effort to compile and standardize the collected data sets that will facilitate their reuse in further studies of the changing Arctic Ocean. The

dataset is available at https://doi.org/10.17882/75345 (Massicotte et al., 2020).

## 1 Introduction

The Mackenzie River is the largest source of terrestrial particles entering the Arctic Ocean (see Doxaran et al. (2015) and references therein). During the past decades, temperature rise, permafrost thawing, coastal erosion, and increasing river runoff have contributed to intensifying the export of terrestrial carbon by the Mackenzie River to the

15 Arctic Ocean (e.g. Tank et al. (2016)). Furthermore, the environmental changes currently happening in the Arctic may have profound impacts on the biogeochemical cycling of this exported carbon. On one hand, reduction in sea-ice extent and thickness expose a larger fraction of the ocean surface to higher solar radiations and increase the mineralization of this carbon into atmospheric $CO_2$ through photo-degradation (Miller and Zepp, 1995; Bélanger

et al., 2006). On the other hand, the possible increase in nutrients brought by Arctic rivers may contribute to higher
autotrophic production and sequestration of organic carbon (Tremblay et al., 2014).

Given that these production and removal processes are operating simultaneously, the fate of arctic river carbon transiting toward the Arctic Ocean is not entirely clear. Hence, detailed studies about these processes are needed to determine if the Arctic Ocean will become a biological source or a sink of atmospheric $CO_2$. With regard to this question, the MALINA oceanographic expedition was designed to document and get insights on the stocks and the
processes controlling carbon fluxes in the Mackenzie River and the Beaufort Sea. Specifically, the main objective of the MALINA oceanographic expedition was to determine how (1) primary production, (2) bacterial activity and (3) organic matter photo-oxidation influence carbon fluxes and cycling in the Canadian Beaufort Sea. In this article, we present an overview of an extensive and comprehensive data set acquired from a coordinated international sampling effort conducted in the Mackenzie River and in the Beaufort Sea in August 2009.

## 2   Study area, environmental conditions and sampling strategy

### 2.1   Study area and environmental conditions

The MALINA oceanographic expedition was conducted between 2009-07-30 and 2009-08-25 in the Mackenzie River and the Beaufort Sea systems (Fig. 1). Figure 2 shows an overview of the sea ice conditions that prevailed during the expedition. In Fig. 2A, a true color image from MODIS Terra reveals how the sea ice pack was fragmented toward
the end of the expedition, specially near the 200 meters isobath (identified by the continuous red line). On the shelf, the sea ice concentration was higher at the beginning of the expedition. During the four weeks cruise, the ice concentration gradually decreased toward the north (Fig. 2B).

The Mackenzie River Basin is the largest in northern Canada and covers an area of approximately 1 805 000 $km^2$, which represents around 20% of the total land area of Canada (Abdul Aziz and Burn, 2006). Between 1972 and
2016, the average monthly discharge (recorded at the Arctic Red River station) varied between 3296 and 23241 $m^3$ $s^{-1}$ (shaded area in Fig. 3A). The period of maximum discharge usually occurs at the end of May with decreasing discharge until December, whereas the period of low and stable discharge extends between December and May. During the MALINA oceanographic cruise, the daily discharge varied between 12600 and 15100 $m^3$ $s^{-1}$ (red segment in Fig. 3A, see also Ehn et al. (2019)). Draining a vast watershed, the Mackenzie River annually delivers on average
2100 Gg C $yr^{-1}$ and 1400 Gg C $yr^{-1}$ of particulate organic carbon (POC) and dissolved organic carbon (DOC), respectively, into the Arctic Ocean (Stein and Macdonald, 2004; Raymond et al., 2007). During the expedition conducted onboard the CCGS Amundsen, the air temperature recorded by the foredeck meteorological tower varied between -2 and 11 °C (Fig. 3B). The average air temperature was 3 °C and usually remained above 0 °C.

## 2.2 General sampling strategy

The sampling was conducted over a network of sampling stations organized into seven transects identified with three digits: 100, 200, 300, 400, 500, 600 and 700 (Fig. 1A). Stations were sampled across these seven shelf–basin transects (south-north) to capture the meridional gradient between the estuary and the open ocean (except for transect 100 across the mouth of the Amundsen Gulf). Within each transect, station numbers were listed in descending order from south to north. Because our goal was to sample in open waters, the order in which the transects were visited depended on the ice cover. On 2009-07-20, just before the mission, a relatively large portion of the shelf was still covered by sea ice (Fig. 2B). Soon after the beginning of the cruise, most of the shelf area was ice free. The shelf region was not ice-free before mid-August. The bathymetry at the sampling stations varied between 2 and 1847 m ($394 \pm 512$ m, mean $\pm$ standard deviation). The stations located in the Beaufort Sea were sampled onboard the Canadian research icebreaker *CCGS Amundsen*. Biological, chemical and optical water column sampling was almost always restricted to the first 400 m of the water column during daytime. Deeper profiles for sampling the whole water column and bottom sediment were usually repeated during nighttime at the same stations. Sediment sampling for fauna and biogeochemistry was conducted at eight stations (110, 140, 235, 260, 345, 390, 680, 690). Two transects (600 and 300) were extended to very shallow waters on the shelf and sampled from either a zodiac or a barge (the bathymetry profiles are shown in Fig. 1B). In the context of this data paper, these two transects were chosen to present an overview of the principal variables measured during the MALINA campaign. A summary of the various sampling strategies is presented below.

## 2.3 CTD and rosette deployment

Onboard the *CCGS Amundsen*, a General Oceanic rosette equipped with a CTD (Seabird SBE-911+) was deployed at each sampling station (Fig. 1). The rosette was equipped with twenty-four 12-L Niskin bottles. The rosette was also equipped with a transmissometer sensor (WetLabs), a PAR sensor (Biospherical), an oxygen sensor (SBE-43), a pH sensor (SBE-18), a nitrate sensor (Satlantic ISUS), a fluorometer (Sea Point) and an altimeter (Benthos). A surface PAR (Biospherical) was also installed on the roof of the rosette control laboratory. A UVP5 (Underwater Vision Profiler, Hydroptics) was also mounted on the rosette frame providing size and abundance of particles above 200 μm and plankton above 700 μm. The Rosette data processing and quality control are described in detail in Guillot and Gratton (2010). Data processing included the following steps: validation of the calibration coefficients, conversion of data to physical units, alignment correction and extraction of useless data. Oxygen sensor calibration was done using Winkler titrations and salinity data were compared with water samples analyzed with a Guideline 8400B Autosal. The quality control tests were based on the International Oceanographic Commission suggested procedures and the UNESCO's algorithm standards (Commission of the European Community, 1993). The recorded data were averaged every decibar. On August $5^{th}$, the pH sensor was replaced by a chromophoric dissolved organic matter (CDOM) fluorometer (Excitation: 350-460 nm/emission 550 nm HW 40 nm; Dr. Haardt Optik Mikroelektronik). The

rosette depth range was restricted to the first 1000 m when carrying the pH, PAR and nitrate sensors because of their rating.

## 2.4 Sediment sampling

Surface sediments were sampled using an USNEL box corer (50 x 50 x 40 cm). Box cores with undisturbed surface were subsampled for (a) lipids (Rontani et al., 2012b), and isotopic signature of lipid biomarkers (Tolosa et al., 2013), stable isotopes (C, N) and manganese and iron oxides (Link et al., 2013) in the 1 cm surface layer, (b) sediment pigments profiles down to 8 cm, and (c) fluxes at the sediment-water interface using on-board microcosms incubations on subcores (10 cm diameter, 20 cm deep) (Link et al., 2013, 2019). At three stations (140, 345, 390), macrofauna

abundance and diversity were determined from sieved and conserved samples (Link et al., 2019). Samples for (a)-(b) were stored frozen until analysis in the respective home labs.

## 3 Data quality control and data processing

Different quality control procedures were adopted to ensure the integrity of the data. First, the raw data were visually screened to eliminate errors originating from the measurement devices, including sensors (systematic or

95 random) and errors inherent from measurement procedures and methods. Statistical summaries such as average, standard deviation and range were computed to detect and remove anomalous values in the data. Then, data were checked for duplicates and remaining outliers. The complete list of variables is presented in Table 1.

## 4 Data description: an overview

The following sections present an overview of a subset of selected variables from the water column. For these

100 selected variables, a brief description of the data collection methods is presented along with general results.

## 4.1 Water masses distribution

According to previous studies (Carmack et al., 1989; Macdonald et al., 1989), five main source-water types can be distinguished in the southeastern Beaufort Sea : (1) meteoric water (MW, Mackenzie River plus precipitation), (2) sea-ice meltwater (SIM), (3) winter polar mixed layer (wPML), (4) upper halocline water (UHW, modified Pacific Water

with core salinity of 33.1 PSU), (5) and lower halocline water (LHW, water of Atlantic origin). In this study, we used the optimum multiparameter (OMP) algorithm to quantify the relative contributions of the different source water types to the observed data (https://omp.geomar.de/). We used salinity, TA, and $\delta^{18}O$ as conservative tracers as well as temperature and $O_2$ concentration as non-conservative tracers, to constrain the water mass analysis, following Lansard et al. (2012). Briefly, the method finds the best fitting fraction ($x$) of ($n+1$) source water types that contribute

to the ($n$) observed values of the selected tracers in a parcel of water via a solution of an overdetermined system of

linear equations that minimizes the residual error. Boundary conditions were applied to the method to guarantee that all fractions calculated were positive and that the sum of all fractions was 100% (mass conservation).

During MALINA, the Mackenzie Shelf was for the most part entirely ice-free, and the ice pack was located beyond the shelf break (Fig. 2). The transition zone was characterized by different expanses of drifting sea-ice. Significant contributions of Meteoric Water (> 25%) to the surface mixed layer (SML) were only observed close to the Mackenzie River mouth and on the inner shelf (Fig. 4). A relatively small fraction of sea-ice meltwater was detected beyond the shelf break, mostly along the transect 600. Below the SML, the wPML was the predominant water mass down to 100 m depth. The UHW extends from the interior ocean onto the outer shelf from 120 to 180 m of depth. Relatively high fractions of UHW were also found at 50 m depth along the Mackenzie and Kugmallit Canyons, which are recognized sites of enhanced shelf-break upwelling caused by wind- and ice-driven ocean surface stresses. Below 200 m depth, the LHW with an Atlantic origin was always the prevailing water mass.

## 4.2 Temperature and salinity from the CTD

Temperature and salinity for the first 100 m of transects 600 and 300, the two transects originating from the Mackenzie River delta, are presented in Fig. 5. They confirm what was found by the water mass analysis (section 4.1): most of the freshwater is coming from the western part of the Mackenzie River delta. This is also in accordance with many studies that documented that during the summer, a combination of ice melting and river runoff was generating a highly stratified surface layer (Carmack and Macdonald, 2002; Forest et al., 2013). The signature of an eddy may be observed at 75 m in the salinity data at 70 °N, approximately 70 km from shore (Fig. 5B).

## 4.3 Underwater bio-optical data

### 4.3.1 Inherent Optical Properties (IOPs) profiling from the ship, the barge and the zodiac

The total, non-water, spectral absorption ($a$), attenuation ($c$) and backscattering coefficients ($b_b$) were measured using a AC9 attenuation and absorption meter and a BB9 scattering meter (WetLabs), a HydroScat-6 and a-Beta sensors (HOBI Labs) either attached to the CTD-Rosette frame onboard the *CCGS Amundsen* or deployed separately from the barge or the Zodiac tender. These devices were using either 10 cm or 25 cm optical path lengths, depending on the turbidity of the water sampled. Detailed information about the deployment and the data processing of the IOP data can be found in Doxaran et al. (2012).

Fig. 6 shows cross-sections of the total absorption and backscattering coefficients at 440 nm ($a(440)$ and $b_b(440)$) derived as $b_b = b_{bp} + b_{bw}$, where $b_{bw}$ is the backscattering coefficient of pure seawater (Morel, 1974). Both $a(440)$ and $b_b(440)$ showed the same patterns along the transects 600 and 300. Close to the estuary, higher absorption (Fig. 6A) and total scattering (Fig. 6B) can be observed at the surface, likely reflecting the important quantities of dissolved and particulate organic matter delivered by the Mackenzie River. Higher values are also observed in transect 600 compared to transect 300, which is further away from the mouth of the Mackenzie River. Both $a(440)$ and $b_b(440)$

decreased rapidly toward higher latitudes where the water of the Mackenzie River mixes with seawater from the Beaufort Sea.

 ### 4.3.2 Particulate and CDOM absorption

Chromophoric dissolved organic matter absorption ($a_{CDOM}$) was measured from water samples filtered with 0.2 μm GHP filters (Acrodisc Inc.), using an UltraPath (World Precision Instruments Inc.) between 200 and 735 nm. In most cases, a 2 meters optical path length was used for the measurement, except for coastal waters near the Mackenzie River mouth (Fig. 1) where a 0.1 meters optical path length was used. Particulate absorption (ap) was measured using a filter-pad technique modified from Röttgers and Gehnke (2012). Briefly, sea-water was filtered through a 25 mm Whatman GF/F (glass-fiber filters) less than 3 h after sampling. Filters were placed in the center of a 150 mm integrating sphere equipped with a handmade Spectralon filter holder. The spectral optical density ($OD(\lambda)$) of the particles retained on the filter was then measured using a PerkinElmer Lambda-19 spectrophotometer, from 300 to 800 nm at 1 nm resolution. More details about particulate and dissolved absorption measurements can be found in Röttgers and Gehnke (2012), Bélanger et al. (2013) and Matsuoka et al. (2012).

Examples of $a_{CDOM}$ spectra measured at the surface for the northernmost and the southernmost stations of transects 600 and 300 are presented in Fig. 7A. The marked influence of the organic matter of terrestrial origin can be observed for the stations located at the mouth of the Mackenzie River (697 and 398). Because the organic matter delivered by the river is highly humic and colored, the absorption at 254 nm was approximately 15 times higher at the southern shelf stations for both transects compared to the northern stations (620 and 320). Likewise, the specific UV absorbance of dissolved organic carbon at 254 nm (SUVA$_{254}$), a metric commonly used as a proxy for assessing both chemical (Weishaar et al., 2003; Westerhoff et al., 2004) and biological reactivity (Berggren et al., 2009; Asmala et al., 2013) of the DOM pool in natural aquatic ecosystems, decreased rapidly along the south-north gradient in both transects 600 and 300 (Fig. 7C). This observation is in accordance with a previous study that showed that SUVA$_{254}$ was higher in inland ecosystems due to elevated lateral connectivity with surrounding terrestrial landscape and organic matter inputs from the tributaries (Massicotte et al., 2017). The decrease in SUVA$_{254}$ toward north stations (Fig. 7C) suggests that terrestrially-derived DOM transiting toward the ocean is gradually degraded into smaller and more refractory molecules.

Particulate absorption spectra ($a_p$) for the northernmost and the southernmost stations of transects 600 and 300 are presented in Fig. 7B. Particulate absorption at the stations located in the estuary (697 and 398) was much higher than that measured at the open water stations (620 and 320). For instance $a_p(443)$ measured at stations 620 (0.03 m$^{-1}$) and 697 (8.62 m$^{-1}$), the northernmost and the southernmost stations of transects at the mouth of the Mackenzie River, shows that ap decreases rapidly along the latitudinal axes. This can be possibly explained because the drained organic and inorganic material from the surrounding landscape of the Mackenzie's watershed is degraded or sediment rapidly as it is transferred to the ocean.

### 4.3.3   Other optical measurements and radiometric quantities

Other optical instruments were attached to the rosette sampler. These include a transmissometer (Wetlabs C-Star, path 25 cm) for beam attenuation measurement, a chlorophyll fluorometer (SeaPoint) and a CDOM fluorometer (Optic & Mikro-elektonik, Germany, see Amon et al. (2003)). Additionally, a LISST-100X (Laser In Situ Scattering and Transmissometry, Sequoia Scientific) was attached to the rosette and provided beam attenuation (532 nm) and forward light scattering measurements at 32 angles from which particle size distribution was estimated. Various optical measurements were also made in the laboratory to determine other IOPs. These include the absorption of colored dissolved ($a_{CDOM}$) and particulate ($a_p$) organic matter, the absorption coefficients of non-algal particles ($a_{NAP}$) and phytoplankton ($a_{phi}$). Apparent optical properties (AOPs) measurements included light transmittance ($T$), photosynthetically available radiation (PAR), downward irradiance ($E_d$), upwelled radiance ($L_u$) and global solar irradiance ($E_s$). The latter three radiometric quantities were measured simultaneously using a Compact-Optical Profiling System (C-OPS) manufactured by Biospherical Instruments Inc. (San Diego, California) that was deployed during MALINA Leg 2b. The principal data products obtained from the C-OPS data were the diffuse attenuation coefficient ($K_d$) plus the water-leaving radiance ($L_W$) including all normalized forms. Detailed methodology and results derived from C-OPS measurements can be found in Doxaran et al. (2012), Antoine et al. (2013), Bélanger et al. (2013) and Hooker et al. (2013).

### 4.4   Nutrients

Samples for nitrate, nitrite, soluble reactive phosphorus and silicate determination were collected into 20 mL polyethylene flasks, immediately poisoned with mercuric chloride (Kirkwood, 1992), and stored for subsequent laboratory analysis according to Raimbault et al. (1990) and Aminot and Kérouel (2007). Ammonium concentrations (40 mL collected into 60 mL polycarbonate tubes) were measured onboard using the sensitive method of Holmes et al. (1999) having a detection limit of 5 nmol L$^{-1}$. Samples for organic matter determination were collected into 50-mL Glass Schott bottles, immediately acidified with 100 µl of 0.5 N $H_2SO_4$ and stored in the dark at 5 °C. Dissolved organic carbon (DOC), dissolved organic nitrogen (DON) and dissolved organic phosphorus (DOP) were determined at the laboratory using the wet-oxidation procedure according to Raimbault et al. (1999a).

Nitrate levels were always very low at the surface, with concentration generally lower than 0.01 µmol L$^{-1}$, except in the Mackenzie plume (Fig. 8). It is interesting to note that nitrate was never entirely depleted, and some traces (0.005 to 0.01 µmol L$^{-1}$) were always detectable in surface waters (Fig. 8A). Ammonium distribution showed the same pattern. Even if concentrations were very low (generally < 0.03 µmol L$^{-1}$), this nutrient, like nitrate, was always detected, suggesting that in situ sources of nitrate and ammonium exist offshore, certainly due to biological processes. Phosphate concentrations showed the opposite distribution (Fig. 8B). Despite nitrogen depletion, surface waters were always phosphate replete. The highest concentrations, around 0.5 µmol L$^{-1}$, were observed far from Mackenzie's mouth, revealing a clear west-east gradient. The silicate distribution was similar to that of nitrate. But

Surface waters were always silicate-repleted with concentrations largely above the detection limit (> 4 µmol L$^{-1}$). The impact of the Mackenzie River was clear, close to the coast for inorganic nutrients and farther offshore for dissolved organic nutrients. A quarter of the estimated annual nutrient supply by the Mackenzie River occurred during July-August. The supply of DON was eight times larger than that of nitrate-N. By contrast, the amount of DOP supplied was only 2.5 times higher than the amount of phosphate (Tremblay et al., 2014). The Mackenzie River enriched the western Canadian Beaufort Shelf with inorganic and organic N, potentially supporting most of the primary production, but not with phosphate or ammonium. Large deliveries of N relative to P by rivers relax coastal communities from N limitation, allowing them to tap into the excess P originating from the Pacific Ocean. Then, river inputs locally rectified the strong regional deficit of inorganic N, i.e. negative N* (Tremblay et al., 2014).

## 4.5   Dissolved Organic Carbon (DOC), Total Dissolved Nitrogen (TDN), Total Hydrolyzable Amino Acids (THAA), and Total Dissolved Lignin Phenols (TDLP$_9$)

Water samples were collected at selected stations and water masses for analyses of dissolved organic carbon (DOC), total dissolved nitrogen (TDN), total hydrolyzable amino acids (THAA), and total dissolved lignin phenols (TDLP$_9$) concentrations. Samples for DOC, TDN, and THAA were gravity-filtered from Niskin bottles using pre-combusted (GF/F) filters (0.7 µm pore size) and stored frozen (-20 °C) immediately after collection in pre-combusted borosilicate glass vials (Shen et al., 2012). Samples for TDLP$_9$ analysis (between 1 and 10 L) were gravity-filtered from Niskin bottles using Whatman Polycap AS cartridges (0.2-µm pore size), acidified to pH between 2.5 and 3 with sulfuric acid and extracted within a few hours using C-18 cartridges (Louchouarn et al., 2000; Fichot et al., 2013). The C-18 cartridges were stored at 4 °C until elution with 30 mL of methanol (HPLC-grade), and the eluent was stored in sealed, pre-combusted glass vials at -20 °C until analysis. DOC and TDN concentrations were measured by high-temperature combustion using a Shimadzu total organic carbon analyzer (TOC-V) equipped with an inline chemiluminescence nitrogen detector and an autosampler (Benner and Strom, 1993). Blanks were negligible and the coefficient of variation between injections of a given sample was typically less than 1%. Analysis of a deep seawater reference standard (University of Miami) every sixth sample was used to check the accuracy and consistency of measured DOC and TDN concentration. Total hydrolyzable amino acids (THAA) were determined as the sum of 18 dissolved amino acids using an Agilent High-Performance Liquid Chromatography system equipped with a fluorescence detector (excitation: 330 nm; emission: 450 nm). Samples (100 µL) of filtered seawater were hydrolyzed with 6 mol L$^{-1}$ hydrochloric acid using a microwave-assisted vapour phase method (Kaiser and Benner, 2005). Free amino acids liberated during the hydrolysis were separated as o-phthaldialdehyde derivatives using a Licrosphere RP18 or Zorbax SB-C18 column (Shen et al., 2012). Detailed methodological information can be found in Fichot et al. (2013) and Shen et al. (2012).

Surface DOC concentrations along the transects 300 and 600 behaved approximately conservatively with salinity, decreasing from 458 µmol L$^{-1}$ in the Mackenzie River end-member (salinity = 0.2 PSU) to 123 µmol L$^{-1}$ at a salinity of 26.69 PSU (Fig. 9A). DOC concentrations in surface waters further decreased to minimum values of $\approx$ 66 µmol L$^{-1}$ offshore (Fichot and Benner, 2011). Concentrations generally increased by a few µmol L$^{-1}$ in the upper halocline

relative to surface values, but then generally decreased with depth, reaching 53-57 $\mu$mol $L^{-1}$ in the lower halocline, and $\approx$ 43-50 $\mu$mol $L^{-1}$ in deep water-masses (depth > 1000 m). Similar to DOC, surface $TDLP_9$ concentrations along transects 600 and 300 behaved approximately conservatively with salinity, decreasing from $\approx$ 93-96 nmol $L^{-1}$ in the Mackenzie River end-member (salinity = 0.2 PSU) to $\approx$ 12 nmol $L^{-1}$ at a salinity of 26.69 PSU (Fig. 9B). Surface concentrations reached minimum values of $\approx$ 2.5 nmol $L^{-1}$ offshore (Fichot et al., 2016). $TDLP_9$ concentrations generally decreased with depth, reaching minimum values of < 1.5 nmol $L^{-1}$ below the halocline. Surface concentrations of THAA along the transects 600 and 300 decreased from 576 nmol $L^{-1}$ in the Mackenzie River end-member (salinity = 0.2 PSU) to 317 nmol $L^{-1}$ at a salinity of 26.69 PSU (Fig. 9C). Unlike DOC and $TDLP_9$, concentrations of THAA did not follow a conservative mixing line along the salinity gradient. Elevated concentrations of THAA were observed in mid-salinity waters in both transects, suggesting plankton production in these regions. In comparison, THAA concentrations in the slope and basin waters were lower and decreased with depth, reaching minimal values of $\approx$ 70 nmol $L^{-1}$ below the halocline (Shen et al., 2012).

## 4.6  Pigments

Water samples (volumes between 0.25 L and 2.27 L) were filtered through glass fibre GF/F filters (25 mm diameter, particle retention size 0.7 $\mu$m). They were immediately frozen at -80 °C, transported in liquid nitrogen, then stored at -80 °C until analysis on land. Samples were extracted in 3 mL HPLC-grade methanol for two hours minimum. After sonication, the clarified extracts were injected (within 24 hours) onto a reversed-phase C8 Zorbax Eclipse column (dimension: 3 x 150 mm, 3.5 $\mu$m pore size). The instrumentation comprised an Agilent Technologies 1100 series HPLC system with diode array detection at 450, 667 and 770 nm of phytoplankton pigments (carotenoids, chlorophylls *a*, *b*, *c* and bacteriochlorophyll-*a*). A total of 22 pigments were analyzed and quantified. Details of the HPLC analytical procedure can be found in Ras et al. (2008).

As illustrated in Fig. 10, the phytoplankton biomass, indicated by total chlorophyll *a* concentrations, was the highest at the coast (up to 3.5 mg $m^{-3}$), decreasing offshore (to about 0.010 mg $m^{-3}$) with the formation of a Sub-surface Chlorophyll Maximum (SCM) around 60 m. In terms of biomass integrated over the sampled depth, values range from 6.2 and 8.9 mg $m^{-2}$ at the coast to 14.3 and 13.2 mg $m^{-2}$ offshore for transects 300 and 600, respectively. In general, the most predominant accessory pigment was fucoxanthin, indicating that diatoms constitute a large proportion of the phytoplankton assemblage. However, in offshore waters and around the SCM, 19′-hexanoyloxyfucoxanthin concentrations were equivalent or sometimes higher than fucoxanthin, suggesting that, in these waters, haptophytes can predominate over diatoms. Other pigments such as chlorophyll *b* and prasinoxanthin, suggest the presence of green algae, and probably micromonas-type cells, especially in coastal waters and at the surface. For more detailed information, see Coupel et al. (2015) who used this dataset applied to the CHEMTAX (CHEMical TAXonomy) chemotaxonomic tool to assess the distribution of phytoplankton communities.

## 4.7 Phytoplankton abundance and diversity

The abundance of the eukaryotic pico- and nano-phytoplankton was measured by flow cytometry onboard the Amundsen with a FACS Aria Instrument (Becton Dickinson, San Jose, CA, USA) following the protocol of Marie et al. (1999).

In transect 300 and 600 (Fig. 11), the abundance of pico- and nano-phytoplankton reached maximal values around 5000 and 3000 cells mL$^{-1}$, respectively. On transect 600, pico-eukaryotes higher abundances were restricted to the surface layer with a 5 to 10-fold drop at 30 m. In contrast, nano-eukaryotes formed clear deep maxima, especially at stations 610 and 680. On transect 300, pico-eukaryotes were also abundant in the surface at the more off-shore stations. Still, they decreased sharply near-shore, while nano-eukaryotes' highest concentrations were near the river mouth, linked to high diatom concentrations (Balzano et al., 2012). The composition of eukaryotic phytoplankton was determined with two different approaches. We isolated 164 cultures using a range of techniques (single-cell isolation, serial dilution, flow cytometry sorting) that have been characterized morphologically and genetically (Balzano et al., 2012, 2017) and deposited to the Roscoff Culture Collection (www.roscoff-culture-collection.org). Among these cultures, several new species have been discovered such as the new species of green algae *Mantoniella beaufortii* (Yau et al., 2020) or the diatom *Pseudo-nitzschia arctica* (Percopo et al., 2016), but more await description in particular among *Pelagophyceae*. One of the strains isolated (RCC2488, *Chlamydomonas malina* nomen nudum) has been recently found to be suitable for biotechnology applications (Morales-Sánchez et al., 2020). We also used molecular approaches by sorting pico- and nano-eukaryotic communities and characterizing their taxonomic composition by TRFLP (terminal-restriction fragment length polymorphism) analysis and cloning/sequencing of the 18S ribosomal RNA gene (Balzano et al., 2012). While the pico-phytoplankton was dominated by the species *Micromonas polaris*, the nano-phytoplankton was more diverse and dominated by diatoms mostly represented by *Chaetoceros neogracilis* and *C. gelidus*, with the former mostly present at surface waters and the latter prevailing in the SCM (Balzano et al., 2012). Furthermore, *C. neogracilis* sampled from the Beaufort Sea consists of at least four reproductively isolated genotypes (Balzano et al., 2017). The comparison between the taxonomy of natural communities and isolated cultures (Fig. 12) reveals that although we succeeded at isolating some dominant species in the field such as *M. polaris*, *C. neogracilis* and *C. gelidus* some other important taxa such as the diatom *Fragilariopsis* or the haptophyte *Chrysochromulina* were not recovered.

## 4.8 Carbon fluxes

In the context of climate change, the main objective of the MALINA oceanographic expedition was to determine how (1) primary production, (2) bacterial activity and (3) photo-degradation influence carbon fluxes and cycling of organic matter in the Arctic. In the following sections, we present an overview of these processes in the water column that are detailed in Ortega-Retuerta et al. (2012a), Xie et al. (2012), Tremblay et al. (2014), and refer to Link et al. (2013), Tolosa et al. (2013) and Rontani et al. (2012b) for the related processes at the sediment-water interface.

### 4.8.1  Phytoplankton primary production

At each station, when productivity was quantified, rates of carbon fixation (primary production) were determined
using a $^{13}$C isotopic technique (Raimbault and Garcia, 2008). For this purpose, three 580 mL samples were collected
at minimum sun elevation or before sunrise at 6-7 depths between the surface and the depth where irradiance was
0.3% of the surface value and poured into acid-cleaned polycarbonate flasks. Incubations were carried out immediately following the tracer addition in an on-deck incubator. This consisted of 6-7 opaque boxes, each with associated
neutral and blue screens, allowing around 50%, 25%, 15%, 8%, 4%, 1% and 0.3% light penetration. At five stations,
incubations were also performed in situ on a drifting rig with incubation bottles positioned at the same depth where
samples for on-deck incubations were collected. After 24 h, samples were filtered through pre-combusted (450 °C)
Whatman GF/F filters (25-mm diameter). After filtration, filters were placed into 2 mL glass tubes, dried for 24 h in
a 60 °C oven and stored dry until laboratory analysis. These filters were used to determine the final $^{13}$C enrichment
ratio in the particulate organic matter on an Integra-CN mass spectrometer. Filtrates were poisoned with $HgCl_2$ and
stored to estimate ammonium regeneration and nitrification rates. The isotopic enrichment of particulate organic
matter and dissolved $NH_4^+$ and $NO_3^-$ at the end of incubations were used to calculate net C and N uptake and the
recycling of $NH_4^+$ and $NO_3^-$ (Raimbault et al., 1999b).

Daily rates of primary production at the surface were generally very low across the survey area, ranging from 0.1
mg C m$^{-3}$ d$^{-1}$ offshore to a maximum of 545 mg C m$^{-3}$ d$^{-1}$ in Kugmallit Bay (Fig. 13) associated with the Mackenzie
River discharge (Tremblay et al., 2014). Ammonification and nitrification followed the same coastal-offshore pattern
with rates driving most, if not all, of the $NH_4^+$ and $NO_3^-$ consumption in the surface layer. Primary production was
generally maximum at the surface, but high rates were often observed at depth in the nitracline layer associated
with a chlorophyll maximum. The range of uptake rates of ammonium at the surface generally overlapped with the
range of nitrate uptake rates. Nitrate uptake below the surface amounted to 40–60% of total nitrogen uptake, a
proportion that is approximately twice greater than at the surface (Ardyna et al., 2017).

Nitrification and ammonium regeneration were detectable over the whole water column ranging from 2 to 20
nmol L$^{-1}$ d$^{-1}$. The highest rates were generally located at the base of the euphotic zone, leading to the formation of
subsurface ammonium and nitrite maximum layers. Surface communities and especially the accumulation of large
cells thrived mostly on regenerative $NH_4^+$ and their reliance on $NO_3^-$ increased with depth to reach a maximum in
the subsurface chlorophyll maximum, where substantial levels of primary production occurred (Ardyna et al., 2017).
This is consistent with Ortega-Retuerta et al. (2012a) who reported elevated bacterial abundance and bacterial production rates in association with photoammonification of riverine organic matter (Le Fouest et al., 2013). Nitrification
accounted for a variable and sometimes a large share of the $NO_3^-$ demand, consistent with the persistence of trace
amounts of $NO_3^-$ at the surface. Collectively, the data indicate that the coastal Beaufort Sea is an active regenerative system during summer, probably fuelled by large pools of organic matter brought by rivers. Consequently,
new production was very low and often close to zero in the 0-40 m layer. But high nitrate uptake rates can be ob-

served at depth (Station 135), often associated with high primary production located in the chlorophyll maximum layer being the place of significant new production. The impact of the Mackenzie River on shelf productivity during summer is moderate and associated mostly with localized nutrient recycling in the nearshore estuarine transition zone (Tremblay et al., 2014).

### 4.8.2 Photo-degradation

#### 4.8.2.1 CO and $CO_2$ production from dissolved organic matter

Surface water samples were gravity-filtered upon collection through a pre-cleaned Pall AcroPak 1000 filtration capsule sequentially containing 0.8 and 0.2 μm polyethersulfone membranes. Filtered water was stored in clear-glass bottles at 4 °C in darkness. CO photoproduction rates ($P_{CO}$, nmol $L^{-1}$ $h^{-1}$) were determined aboard the *CCGS Amundsen* immediately after sample collection, whereas $CO_2$ photoproduction rates ($P_{CO2}$, nmol $L^{-1}$ $h^{-1}$) were measured in a land-based laboratory in Rimouski, Québec within three months of sample collection. The sample-pretreatment and irradiation procedures followed those reported previously (Bélanger et al., 2006; Song et al., 2013). Briefly, after minimizing the background CO and $CO_2$ concentrations, samples were transferred into combusted, quartz-windowed cylindrical cells (CO: i.d.: 3.4 cm, length: 11.4 cm; $CO_2$: i.d.: 2.0 cm, length: 14 cm) and irradiated at 4 °C using a SUNTEST XLS+ solar simulator equipped with a 1.5-kW xenon lamp. The radiation emitted from the solar simulator was screened with a Schott long-pass glass filter to remove UV radiation < 295 nm. The irradiations lasted for 10 min to 2 h for CO and 24 to 48 h for $CO_2$. The photon flux reaching the quartz windows of the cells was measured to be 835 μmol $m^{-2}$ $s^{-1}$ for CO and 855 μmol $m^{-2}$ $s^{-1}$ for $CO_2$ over the wavelength range from 280 to 500 nm.

Both $P_{CO2}$ and $P_{CO}$ increased landward, with the difference between the most and least saline samples reaching a factor of $\approx$ 5 along transect 300 and $\approx$ 8 along transect 600 for $P_{CO2}$ and of $\approx$ 7 along transect 600 for $P_{CO}$ (Fig. 14A). This landward increase in $P_{CO2}$ and $P_{CO}$ was due principally to the parallel augmentation in CDOM absorption, as demonstrated by the linear relationships between these two rates with CDOM absorption: $P_{CO2} = 279.1 \times a_{CDOM}(412) - 17.0$ ($R^2 = 0.964, n = 9$) and $P_{CO} = 17.5 \times a_{CDOM}(412) - 4.8$ ($R^2 = 0.966, n = 7$), where $a_{CDOM}(412)$ ($m^{-1}$) is the CDOM absorption coefficient at 412 nm published previously (Song et al., 2013; Taalba et al., 2013). The irradiance-normalized $P_{CO2}/P_{CO}$ ratio gradually decreased landward along transect 600, from 23.5 at station 691 to 16.2 at station 697, suggesting that the near-shore samples were more efficient at CO photoproduction relative to $CO_2$ photoproduction than the shelf samples. The $P_{CO2}/P_{CO}$ ratios at the two stations on transect 300 were, however, similar (18.9 for station 394 and 20.1 for station 396). Combining the $P_{CO2}/P_{CO}$ ratios from both transects arrives at an average ratio of 19.8 ($\pm$ 2.5 SD), with a rather small relative standard deviation of 12.5%.

It should be pointed out that extrapolating the lab-determined $CO_2$ and CO photoproduction rates to the sampling area is practically infeasible due to the very different laboratory and real-environmental conditions. For instance, the water column in the Mackenzie estuary and shelf areas contains large amounts of particles (Doxaran et al., 2012),

which are also optically active, whereas the irradiated samples were particles-free. Furthermore, the photoproduction rates in the water column would decrease rapidly with depth because of the strong light attenuation by CDOM and particles, while the laboratory radiation at best simulated the radiation of the top 1-2 cm layer of the water column even without considering the constant vs. varying irradiance from the solar simulator and natural sunlight, respectively. To estimate the areal photoproduction rates in the water column from lab-derived data often require coupled optical-photochemical modelling that incorporates spectral apparent quantum yields of the photoproduct of interest (Bélanger et al., 2006; Xie et al., 2009; Fichot and Miller, 2010). Using this approach and CO data from the Malina cruise, Song et al. (2013) estimated a yearly-averaged areal CO photoproduction rate of 9.6 µmol $m^{-2}$ $d^{-1}$ in the Mackenzie estuary and shelf areas, which implies a yearly-averaged areal $CO_2$ photoproduction rate of 191.1 µmol $m^{-2}$ $d^{-1}$ based on the average $P_{CO_2}/P_{CO}$ ratio of 19.8 obtained above. Aggregating the $CO_2$ and CO rates gives a total photomineralization rate of 199.7 µmol C $m^{-2}$ $d^{-1}$.

#### 4.8.2.2   Autoxidation of suspended particulate material

Water samples were filtered immediately after collection through a pre-combusted glass fibre filter (Whatman GF/F, 0.7 µm) under a low vacuum. The filters were frozen immediately at -20 °C until analysis and transported to the laboratory. Treatment of the filters involved $NaBH_4$ reduction and classical alkaline hydrolysis (Rontani et al., 2012a). Reduction of labile hydroperoxides to alcohols is essential for estimating the importance of autoxidative degradation in natural samples by gas chromatography-electron impact mass spectrometry (GC-EIMS) (Marchand and Rontani, 2001). Autoxidative degradation of terrigenous particulate organic matter (POM) discharged by the Mackenzie River was monitored thanks to specific oxidation products of sitosterol (main sterol of higher plants) and dehydroabietic acid (a component of conifers).

The autoxidation state of these tracers increases strongly at the offshore stations (Fig. 14B) (reaching 89 and 86% at station 690 and station 380, respectively, in the case of sitosterol, see (Rontani et al., 2014)). These results allowed us to demonstrate that in surface waters of the Beaufort Sea, autoxidation strongly affects vascular plant lipids and probably also the other components of terrestrial OM delivered by the Mackenzie River. Initiation of these abiotic oxidation processes was attributed to the involvement of some enzymes producing radicals (lipoxygenases) present in higher plant debris and whose activity is enhanced at high salinities (Galeron et al., 2018).

#### 4.8.2.3   Bacterial production and respiration

Bacterial production (BP, assessed by [3]H-leucine incubations, $n = 171$), and respiration (BR, assessed by changes in $O_2$ by Winkler titration, $n = 13$), were measured from surface to 200 m at 44 sampling locations. Bacterial production ranged from 8.8 to 7078 µg C $m^{-3}$ $d^{-1}$ and showed a marked decreasing pattern from the mouth of the Mackenzie to the open Beaufort Sea and from the surface to deep waters (Fig. 15). Temperature and labile dissolved organic matter (indicated as dissolved amino acids) controlled BP variability (Ortega-Retuerta et al., 2012a), and the nitrogen

limitation of surface BP during the summer period was demonstrated experimentally (Ortega-Retuerta et al., 2012b). BR ranged from 5500 to 45500 µg C m$^{-3}$ d$^{-1}$, leading to a bacterial growth efficiency of 8% on average. BP and BR were low with respect to lower latitudes but within the range of those in polar ecosystems, suggesting the role of low temperatures driving carbon fluxes through bacteria (Kirchman et al., 2009). Bacterial carbon demand (BP + BR), which averaged 21500 $\pm$ 14900 µg C m$^{-3}$ d$^{-1}$, was higher than primary production in the whole study area, indicating that the Mackenzie River platform and the Beaufort Sea are net heterotrophic during summer. This may suggest a temporal decoupling between carbon fixation and remineralization in the area.

### 4.8.3 Bacterial diversity

Spatial variations in bacterial community structure were explored in surface waters from the Mackenzie River to the open Beaufort Sea ($n = 20$). By using 16S rRNA-based analysis, we investigated both particle-attached (PA, > 3 µm size fraction) and free-living bacteria (FL, size fraction between 3 and 0.2 µm) along a river to open sea transect. Multivariate statistical analysis revealed significant differences in community structure between the river, coastal and open sea waters, mainly driven by salinity, particle loads, chlorophyll *a*, and amino acid concentration (Ortega-Retuerta et al., 2013). Bacterial communities differed between PA and FL fractions only in open sea stations, likely due to the higher organic carbon content in particles with respect to particles from the river and coast, which were enriched in minerals. Alphaproteobacteria dominated in FL open sea samples, while the PA fraction was mainly composed of Gammaproteobacteria, Opitutae (Verrucomicrobia) and Flavobacteria. The coastal and river samples were dominated by Betaproteobacteria, Alphaproteobacteria, and Actinobacteria in both the PA and FL fractions (Fig. 15C). These results highlight the importance of particle quality, a variable that is predicted to change along with global warming, in influencing bacterial community structure, and thus likely altering the biogeochemical cycles that they mediate.

## 5  Conclusions

The comprehensive data set assembled during the MALINA oceanographic cruise has given unique insights on the stocks and the processes controlling carbon fluxes in the Mackenzie River and the Beaufort Sea. In this paper, only a handful of variables has been presented. The reader can find the complete list of measured variables in Table 1, all of which are also fully available in the data repository. The uniqueness and comprehensiveness of this data set offer more opportunities to reuse it for other applications.

## 6  Code and data availability

Metadata and detailed information about measurements can be found in associated MALINA papers presented in Table 1. Data is provided as a collection of comma separated values (CSV) files that are regrouping measurements associated with a particular type of measure. To aid the user to merge these files, there is a lookup table file called

stations.csv that can serve as a table to join all the data together based on date, time, station, cast, depth, longitude and latitude. Additionally, original data provided by all the researchers, as well as additional metadata, are

440 available on the LEFE-CYBER repository (http://www.obs-vlfr.fr/proof/php/malina/x_datalist_1.php?xxop=malina&xxcamp=malina). The processed and tidied version of the data is hosted at SEANOE (SEA scieNtific Open data Edition) under the CC-BY license (https://www.seanoe.org/data/00641/75345/, Massicotte et al. (2020)). The raw UVP5 large particulate data and images are all available from the EcoPart/Ecotaxa website (https://ecotaxa.obs-vlfr.fr/part/). Note that Table 1 also indicates if the measured variables are directly available in the data files or by contacting re-

445 sponsible principal investigators. For specific questions, please contact the principal investigator associated with the data (see Table 1). If more data become available, they will be added to the SEANOE repository. The code used to produce the figures and the analysis presented in this paper is available under the GNU GPLv3 license (https://doi.org/10.5281/zenodo.4518943).

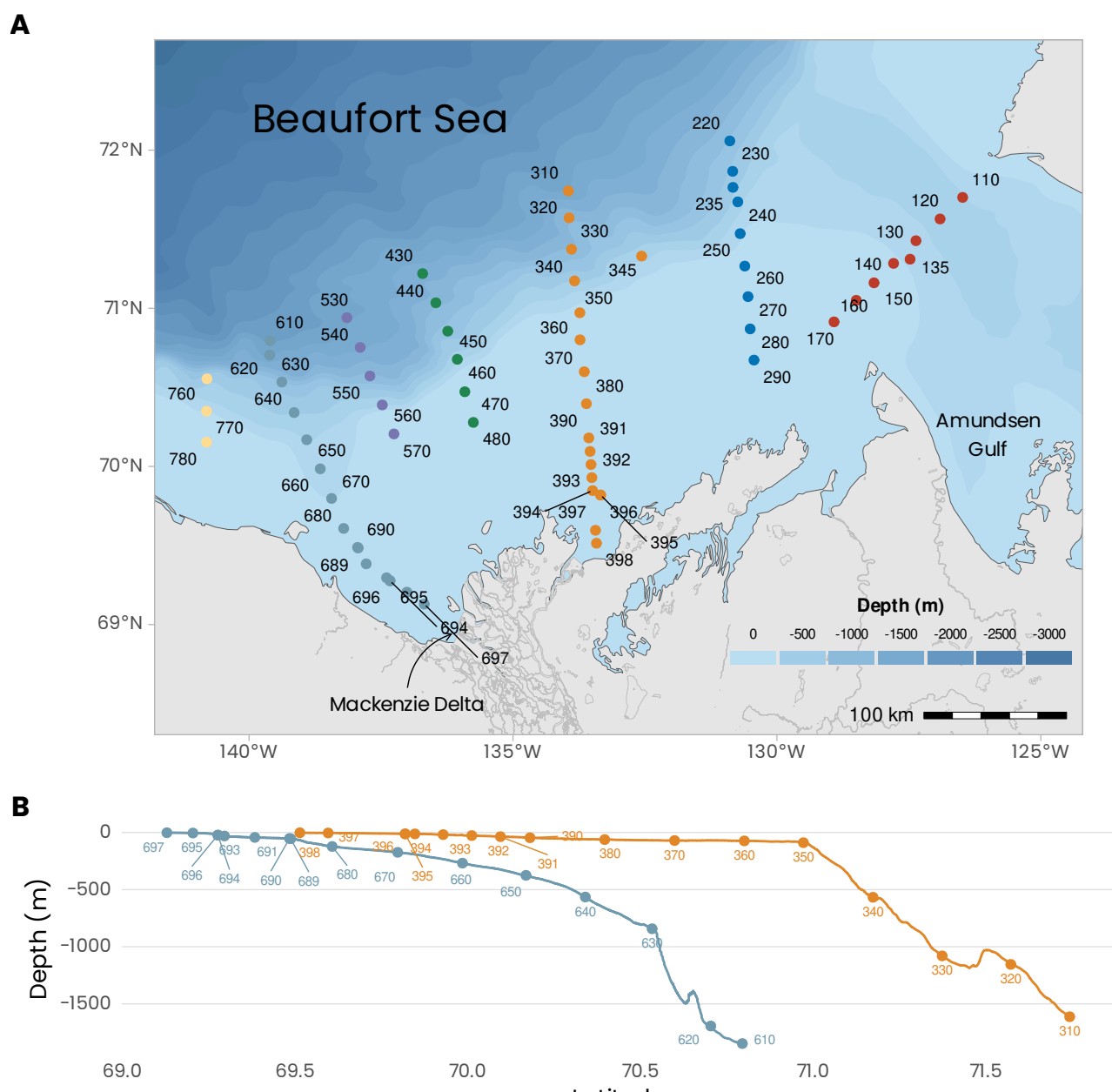

**Figure 1.** (**A**) Localizations of the sampling sites visited during the MALINA 2009 campaign. The colors of the dots represent the seven transects visited during the mission. (**B**) Bathymetric profiles for transects 600 and 300. Bathymetric data from GEBCO (https://download.gebco.net/).

**A**

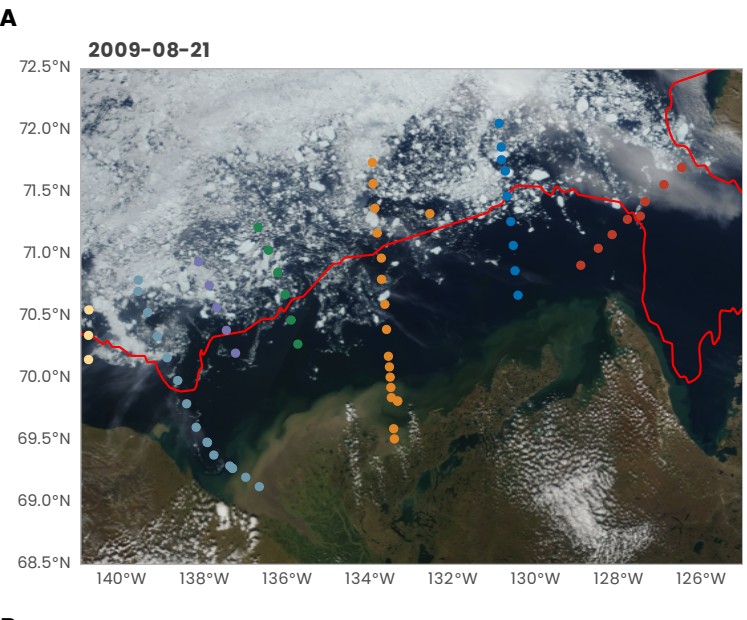

**B**

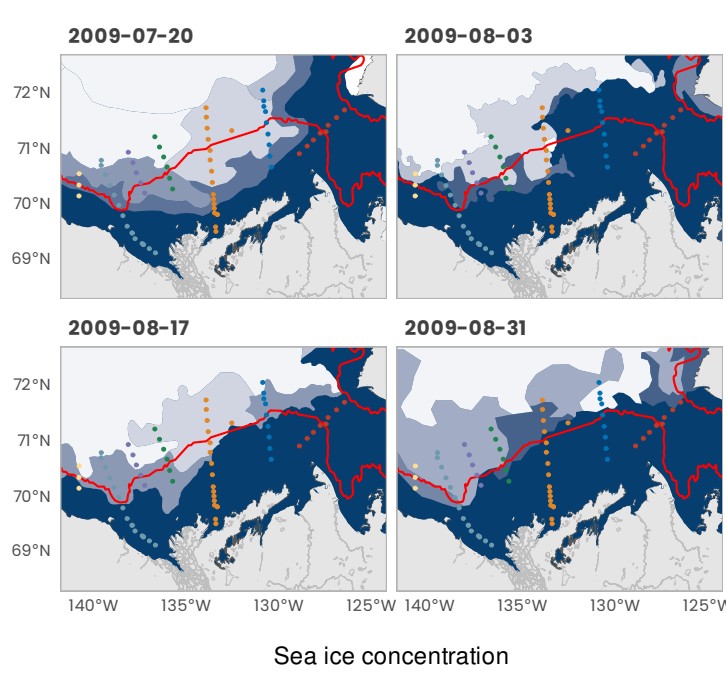

Sea ice concentration

0%   25%   50%   75%   100%

**Figure 2.** (**A**) True color image from MODIS Terra (data from https://wvs.earthdata.nasa.gov). (**B**) Weekly sea ice concentration from the U.S. National Ice Center (U.S. National Ice Center, 2020). The red line shows the 200 meters isobath (data from www.naturalearthdata.com). The dots represent the stations (see Figure 1 for the legend).

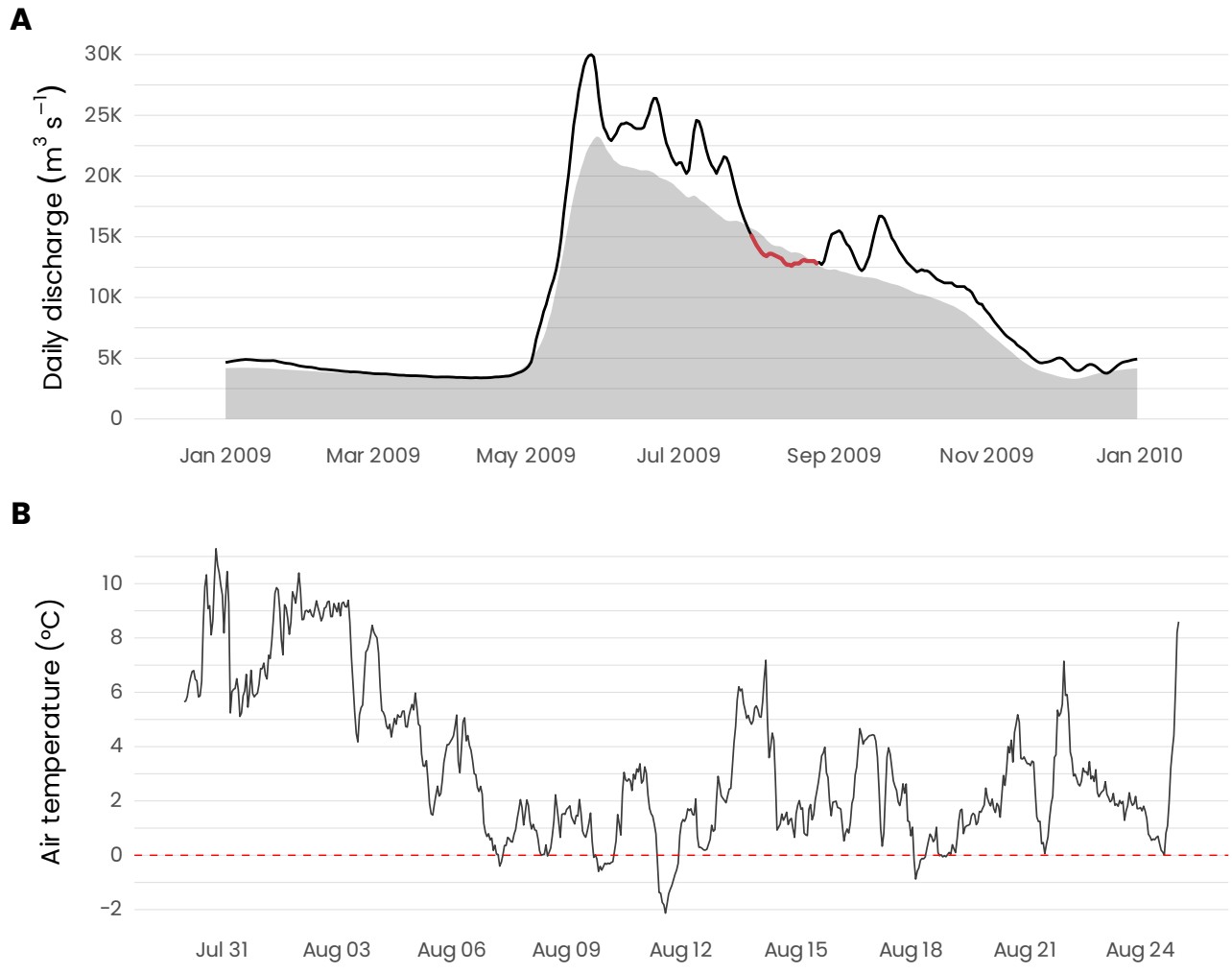

**Figure 3.** (**A**) Daily discharge of the Mackenzie River at the Arctic Red River junction (station 10LC014). The black line corresponds to the 2009 discharge whereas the coloured segment identifies the period of the MALINA campaign. The shaded area is the mean discharge calculated between 1972 and 2016. Discharge data from the Government of Canada (https://wateroffice.ec.gc.ca/search/historical_e.html). (**B**) Hourly air temperature recorded from the Amundsen's foredeck me-teorological tower during the campaign.

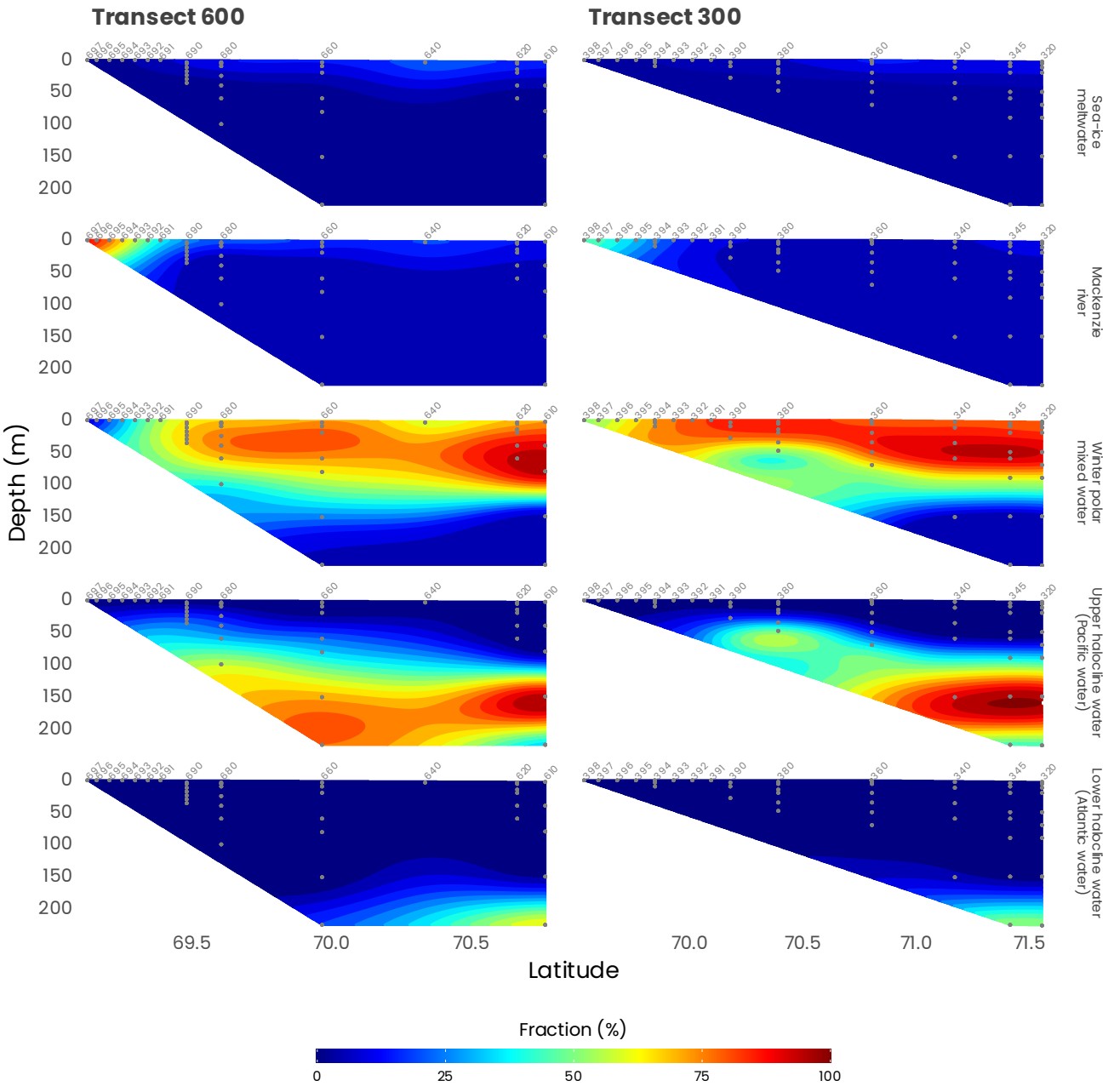

**Figure 4.** Distribution of source water types along transects 600 and 300 (see Fig. 1). Station numbers are identified in light gray on top of each panel.

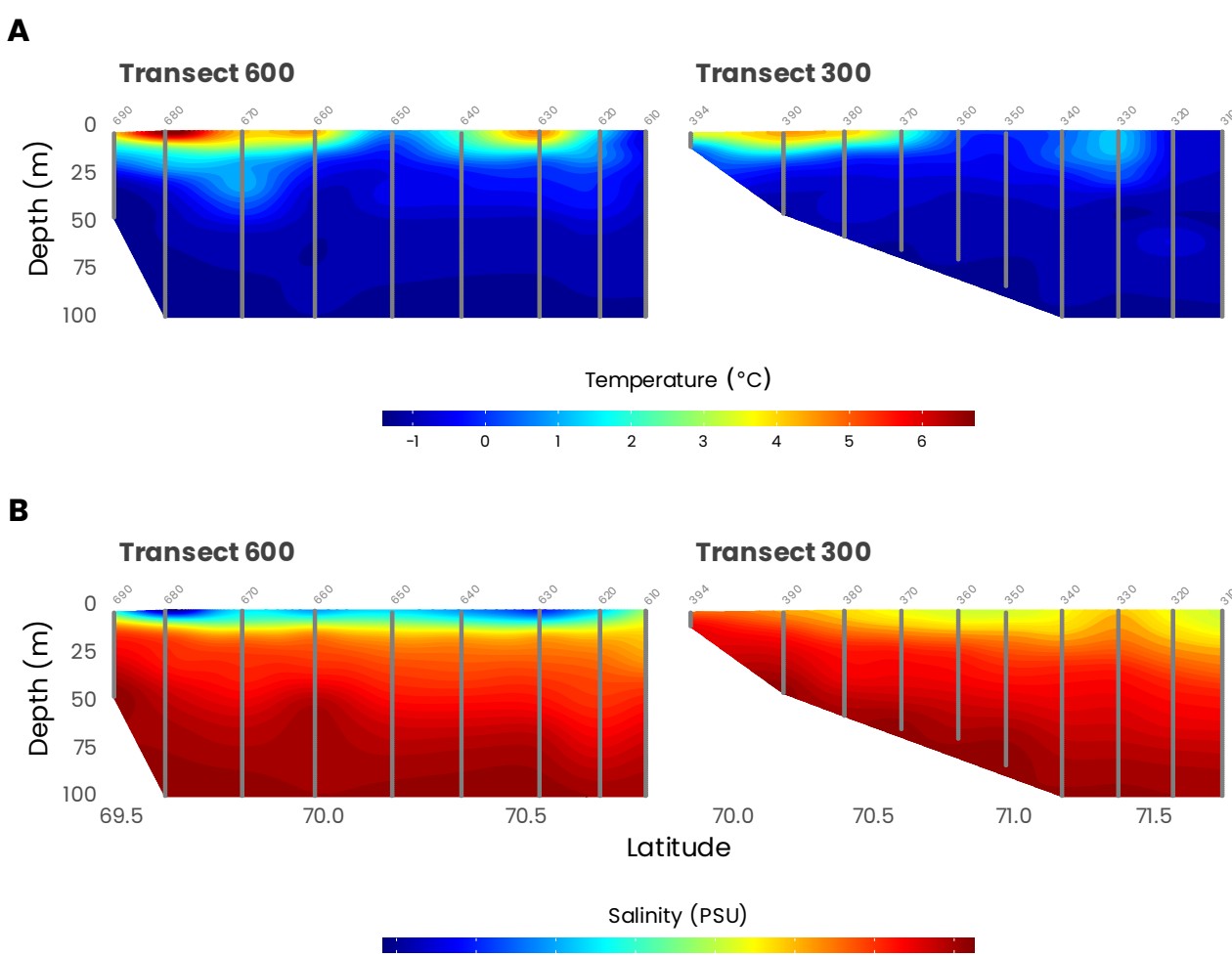

**Figure 5.** Cross-sections of temperature (**A**) and salinity (**B**) measured by the CTD (gray dots) along transects 600 and 300. Station numbers are identified in light gray on top of each panel.

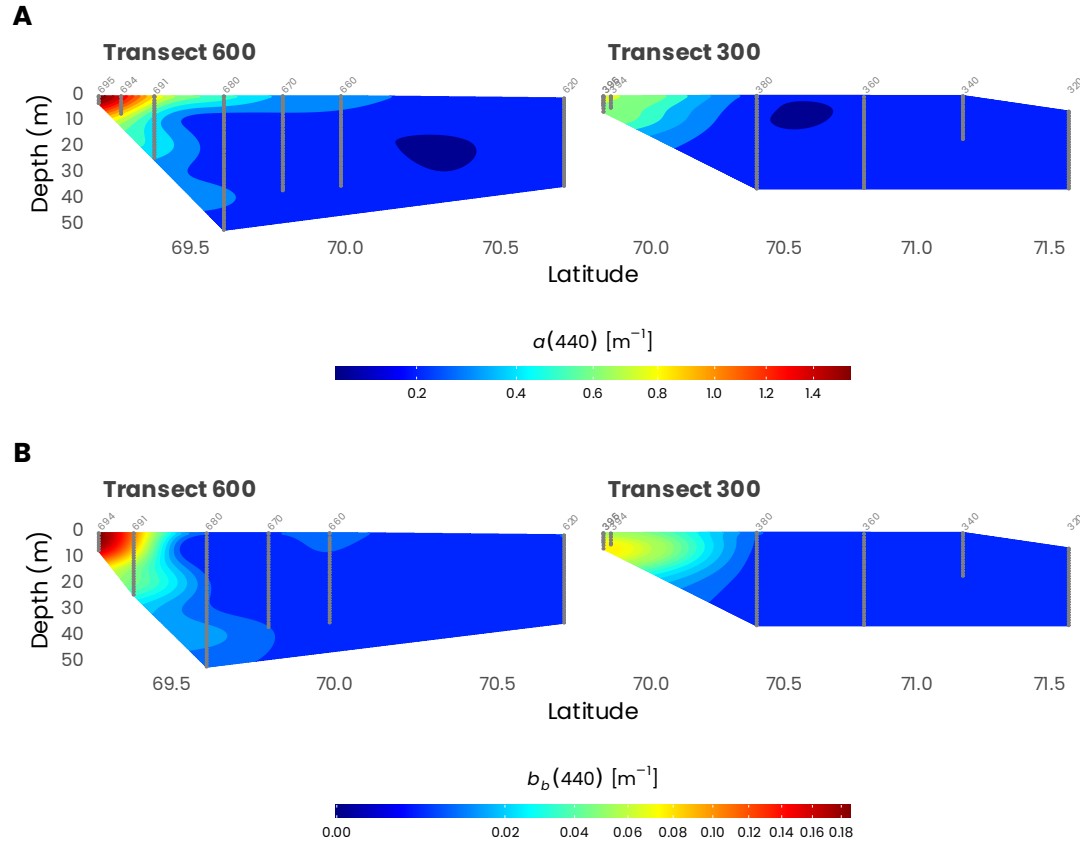

**Figure 6.** Cross-sections of (**A**) absoprtion ($a(440)$) and (**B**) total scattering ($b_b(440)$) measured from the barge at 440 nm with an AC9 and BB9 respectively along transects 600 and 300. Station numbers are identified in light gray on top of each panel. Note that the data has been square-root transformed for the visualization.

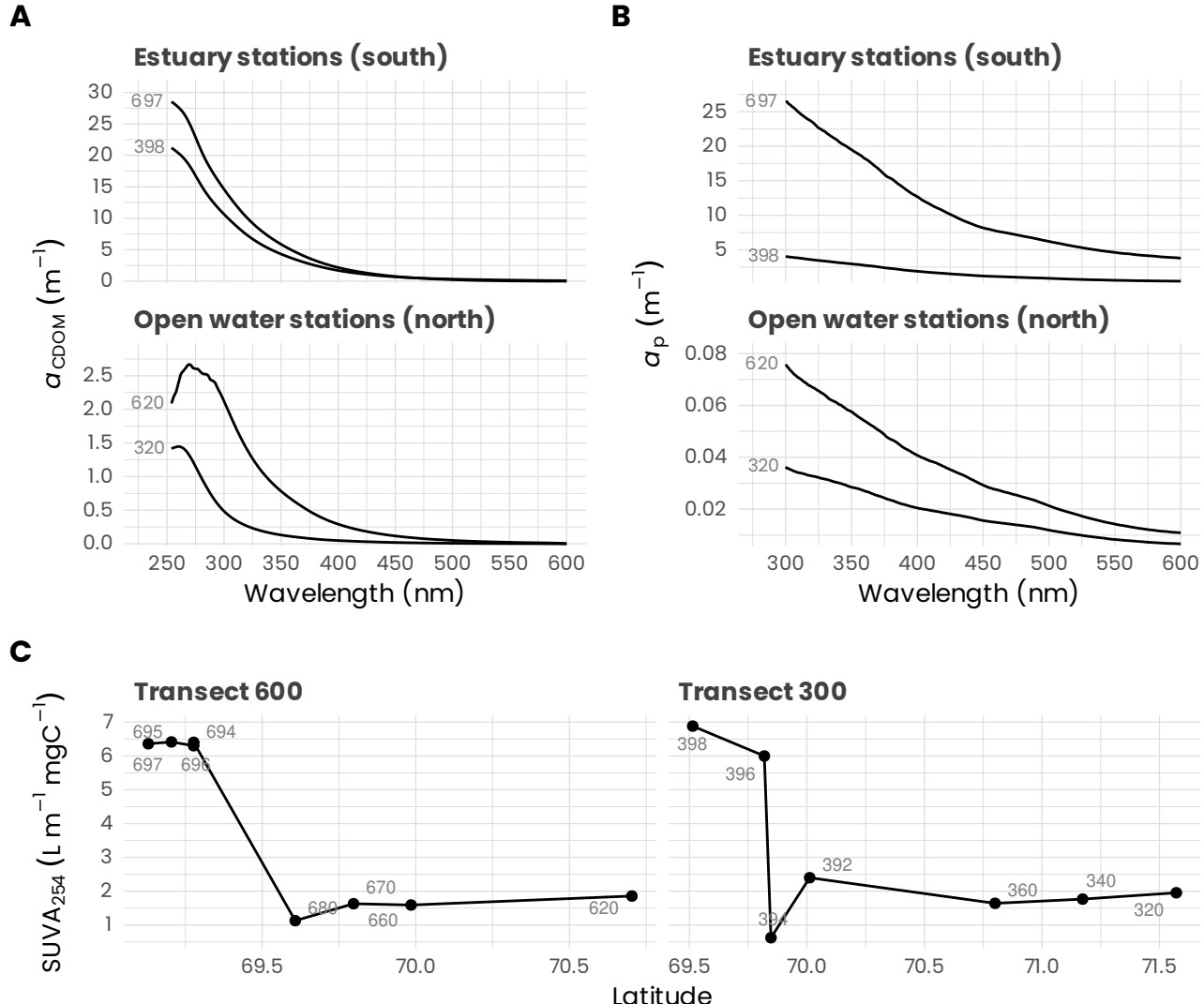

**Figure 7.** (**A**) Absorption spectra between 254 and 600 nm of chromophoric dissolved organic matter ($a_{CDOM}$) measured at the surface for the northern (620, 320) and southern (697, 398) stations of the transects 600 and 300. (**B**) Particulate absorption spectra ($a_p$) measured between 300 and 600 nm measured at the surface for the northernmost and the southernmost stations of the transects 600 and 300. (**C**) Specific UV absorbance at 254 nm (SUVA$_{254}$, i.e. absorption of light at 254 nm per unit of carbon) at surface for stations along transects 600 and 300. Stations are identified in light gray (see Fig. 1 for an overview of the station locations). Note the difference of the y-axes used in panels A and B which highlight the important differences in dissolved and particulate absorption between stations in the estuary and those offshore.

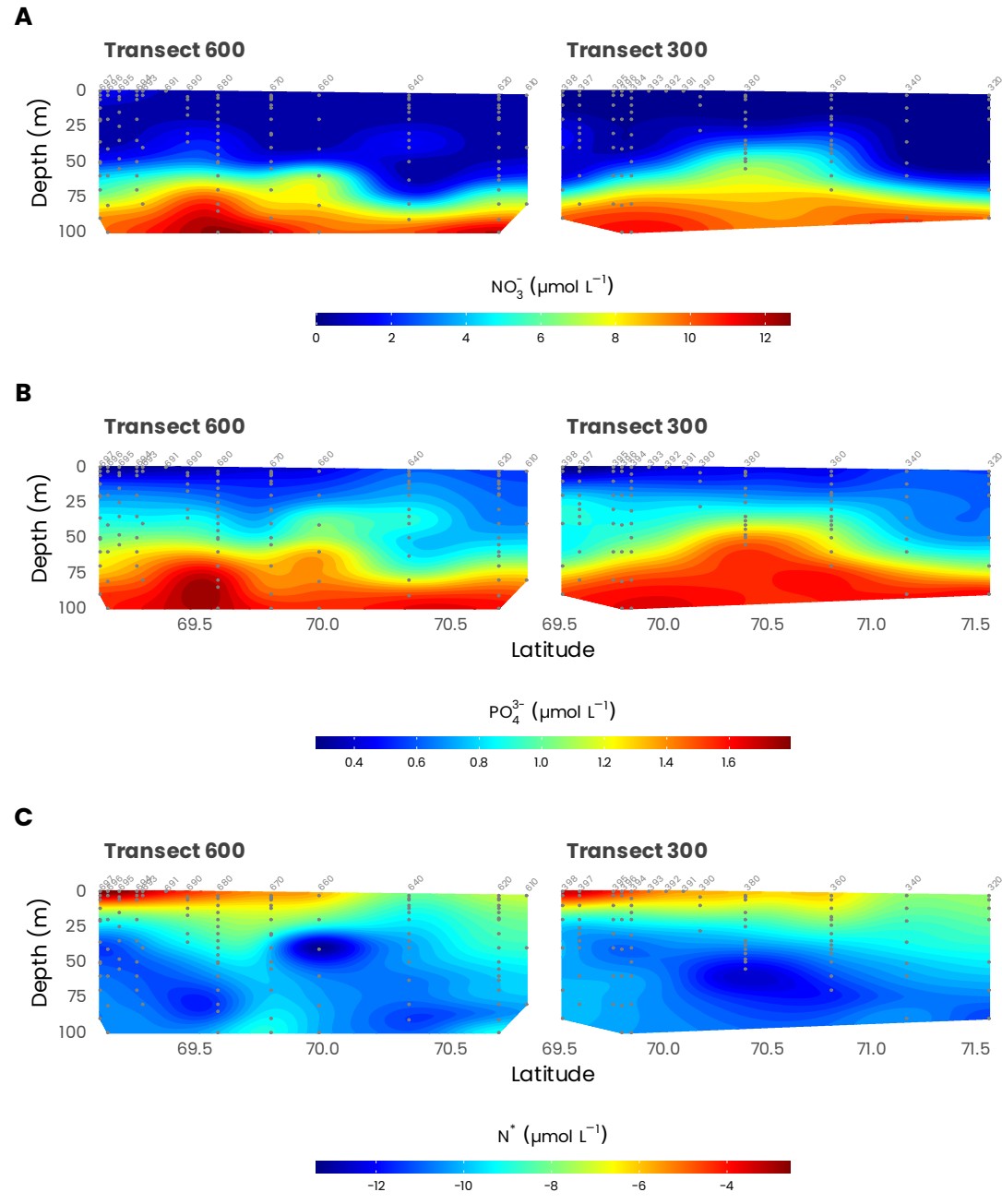

**Figure 8.** Cross-sections of (**A**) $NO_3^-$ and (**B**) $PO_4^{3-}$ measured from Niskin bottles (gray dots) along transects 600 and 300. (**C**) $N^*$ defined as N - rP with r = N/P = 13.1 (see the text for the details). Station numbers are identified in light gray on top of each panel.

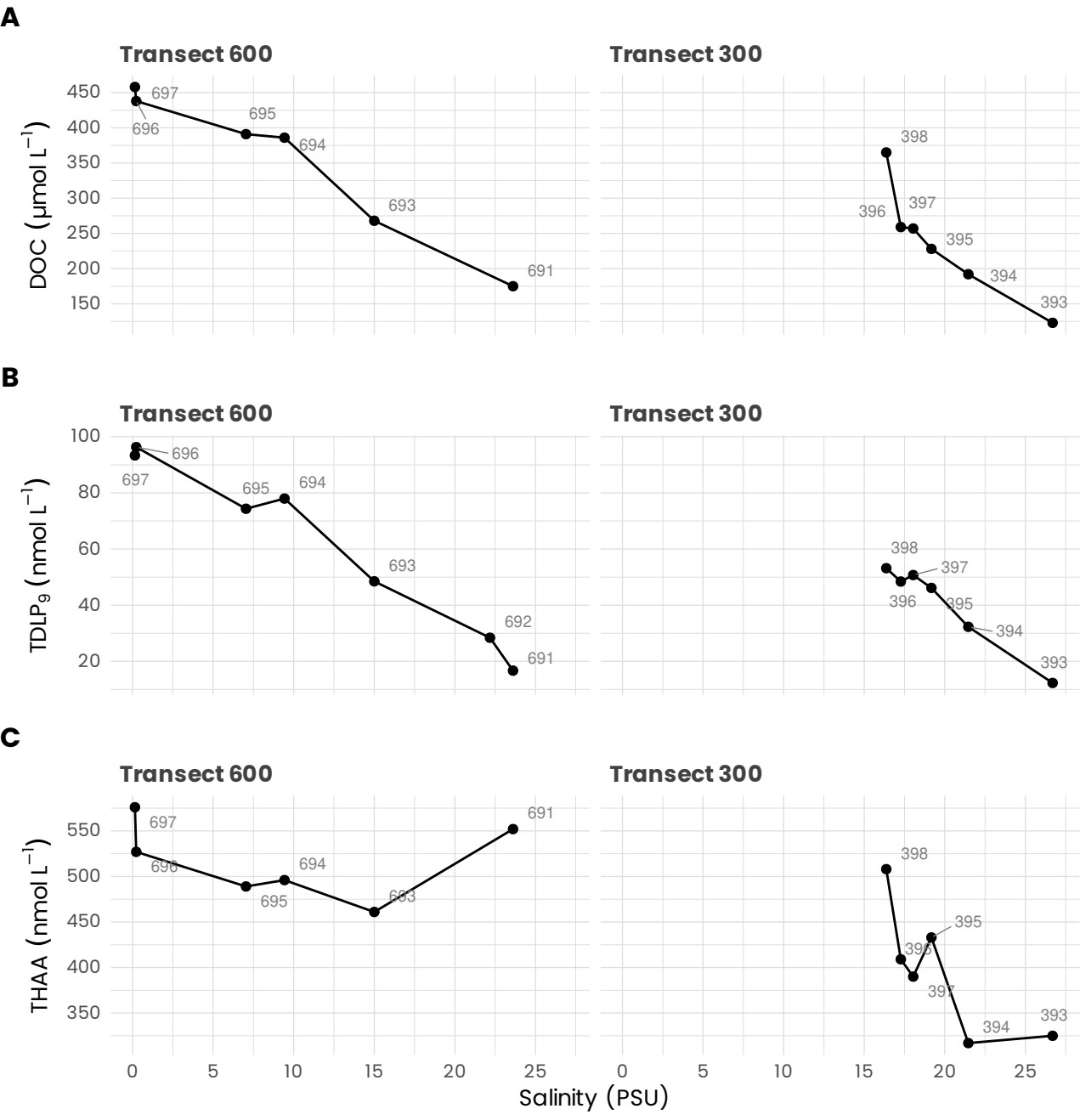

**Figure 9.** Surface concentrations of (**A**) dissolved organic carbon (DOC), (**B**) total dissolved lignin phenols (TDLP$_9$), and (**C**) total hydrolysable amino acids (THAA) measured along transects 600 and 300, and plotted against salinity.

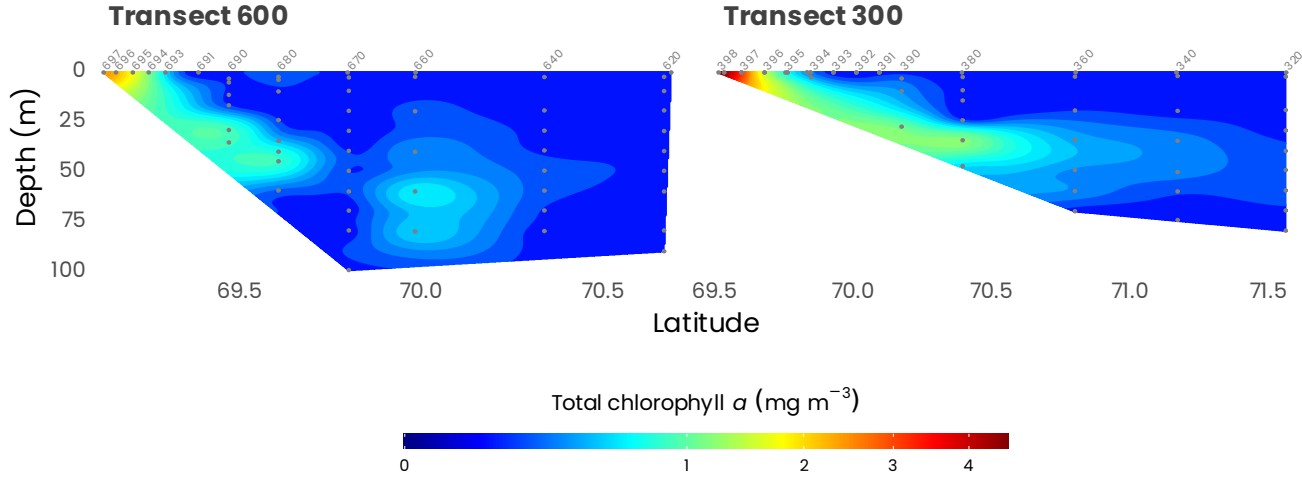

**Figure 10.** Cross-sections of total chlorophyll *a* measured from HPLC (gray dots) along transects 600 and 300. Station numbers are identified in light gray on top of each panel. Note that the data has been square-root transformed for the visualization.

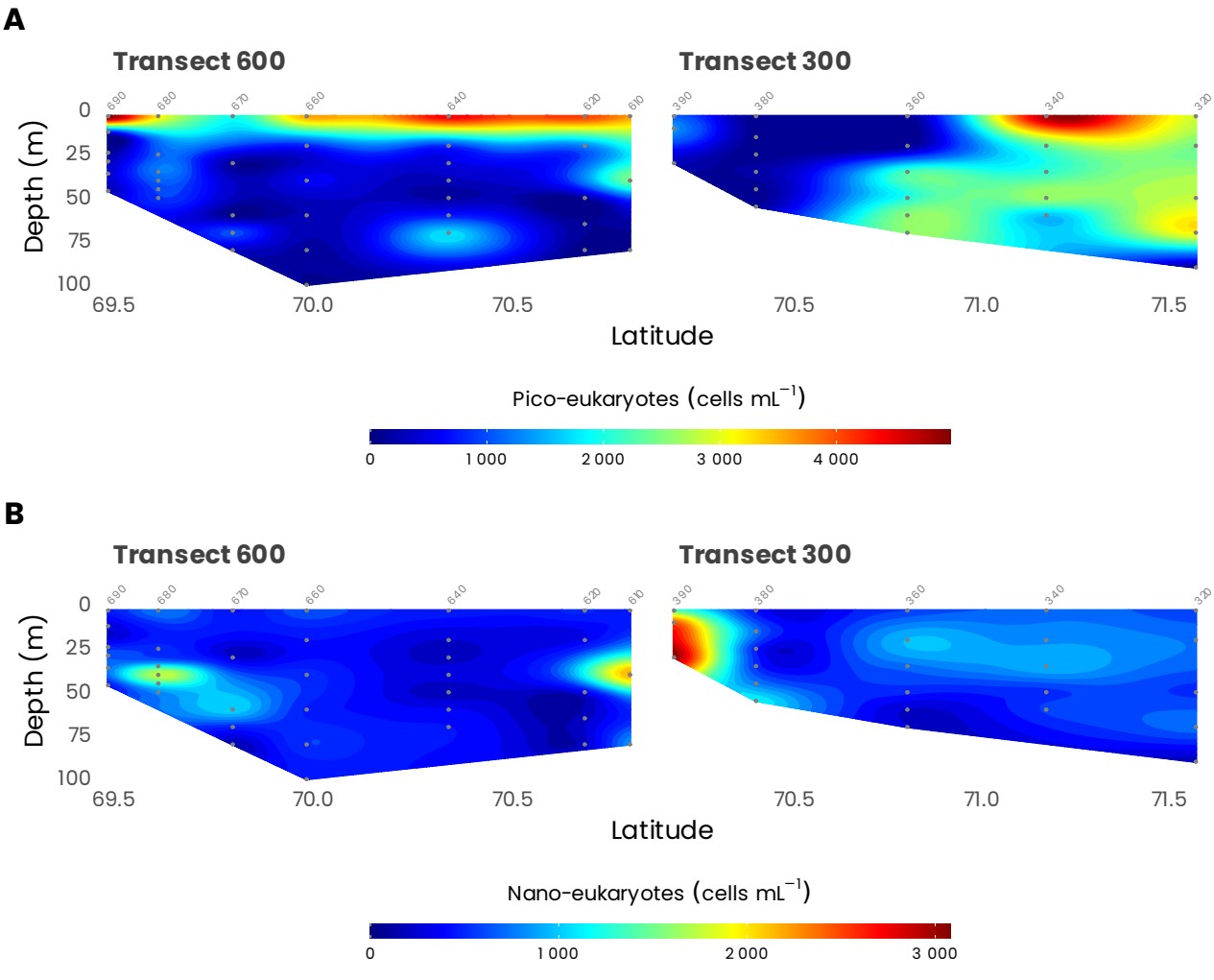

**Figure 11.** Concentrations of photosynthetic (**A**) pico- and (**B**) nano-eukaryotes measured by flow cytometry during the MALINA cruise on transects 600 and 300.

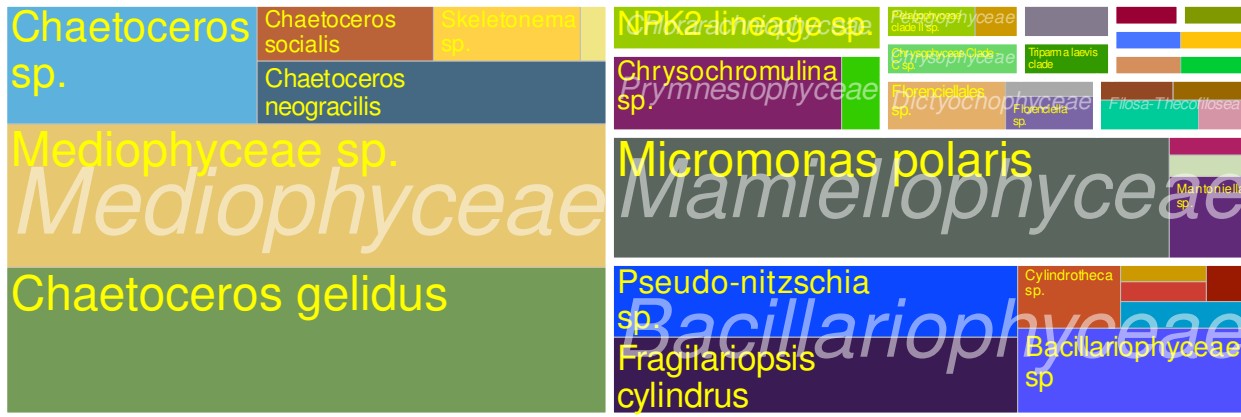

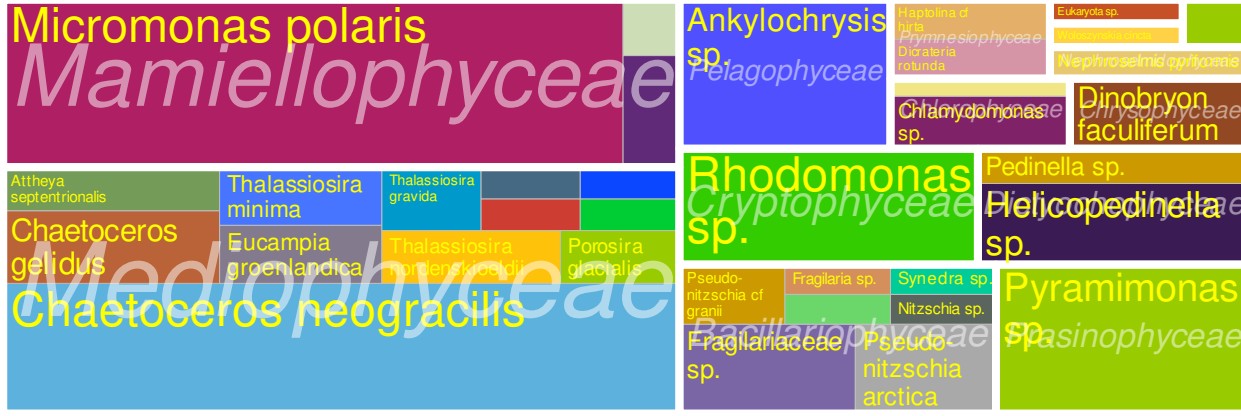

**Figure 12.** (**A**) Taxonomic composition of populations of photosynthetic pico- and nano-eukaryotes sorted flow cytometry from clone library sequences (Balzano et al., 2012). (**B**) Taxonomic composition of cultures of phytoplankton isolated during the MA-LINA cruise (Balzano et al., 2012).

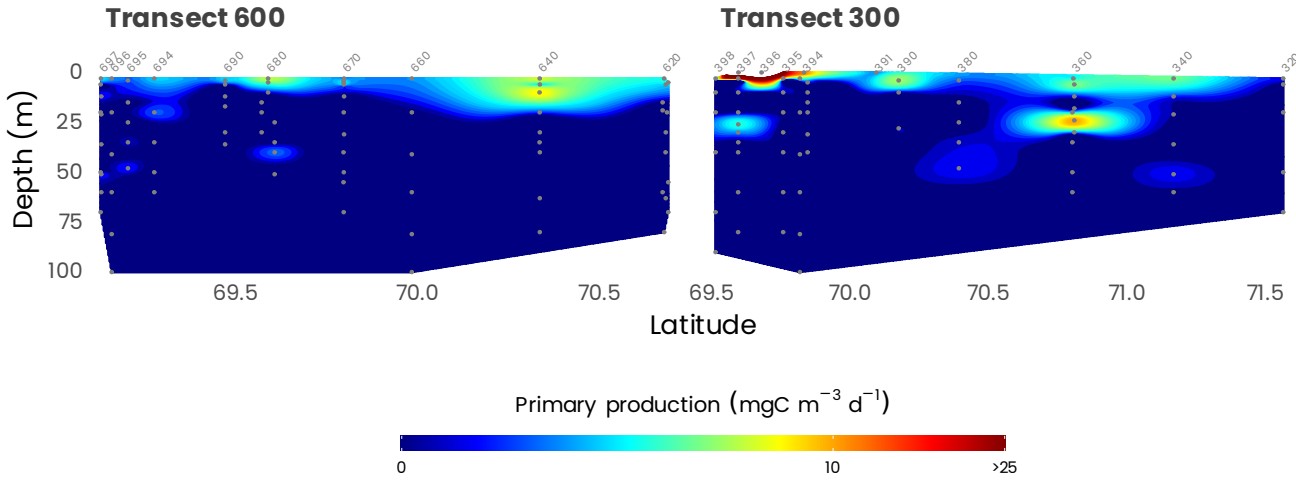

**Figure 13.** Cross-sections of primary production (gray dots) along transects 600 and 300. Station numbers are identified in light gray on top of each panel. Note that the color scale is presented on a log10 scale.

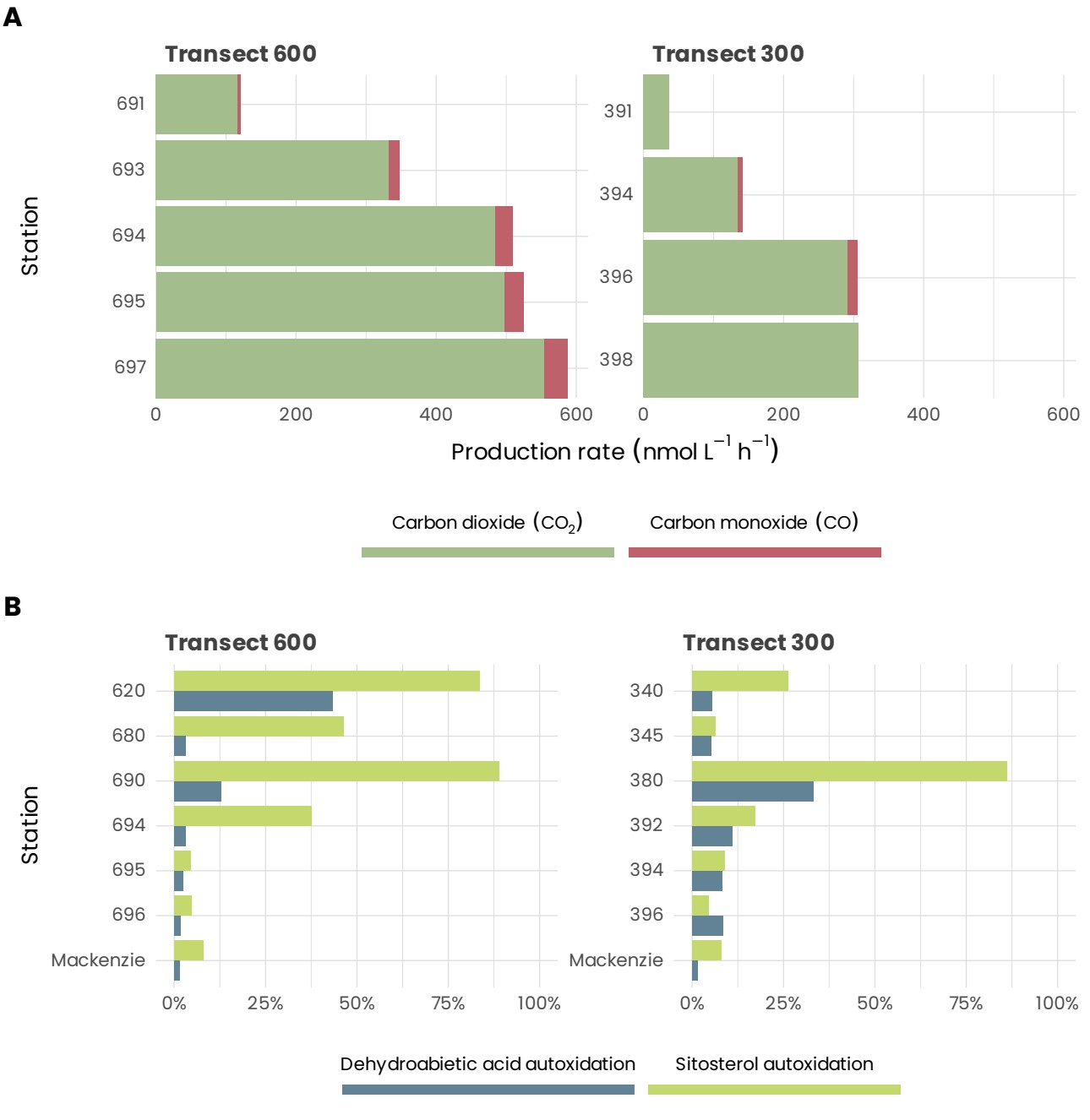

**Figure 14.** (**A**) CO and CO$_2$ production measured at 295 nm at surface for stations of transects 600 and 300. (**B**) Autoxidation of suspended particulate material for stations of transects 600 and 300.

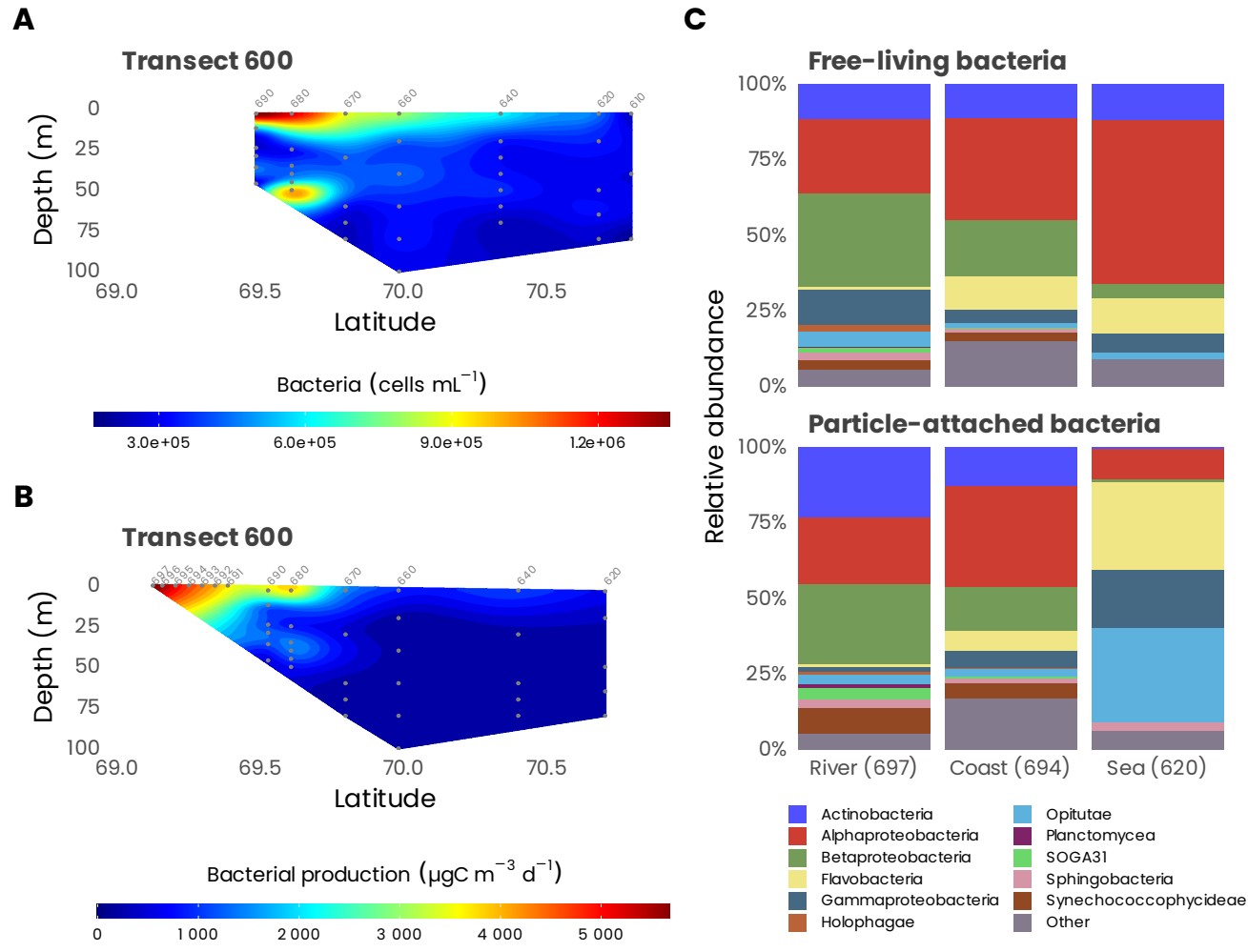

**Figure 15.** (**A**) Cross-sections of bacterial abundance measured from flow cytometry and (**B**) bacterial production measured along transect 600. Station numbers are identified in light gray on top of each panel. (**C**) Cumulative bar charts comparing the relative class abundances in particle-attached (PA) and free-living (FL) for a selected number of samples in transect 600.

Table 1: Parameters measured during the MALINA oceanographic expedition. Parameters are ordered alphabetically.

| Parameters | Method | Sampling | Principal investigators | Included in the data repository | Reference |
|---|---|---|---|---|---|
| [137]Cs datation of core samples | Gamma spectrometer | CASQ corer | Rochon A./ Schmidt | N | 1 |
| [137]Cs datation of core samples | Gamma spectrometry | Box corer | Rochon A./ Schmidt | N | 1 |
| [14]C datation of core samples | Accelerator Mass Spectrometry | Box corer | Rochon A. | N | 1 |
| [14]C datation of core samples | Accelerator Mass Spectrometry | CASQ corer | Rochon A. | N | 1 |
| [15]N-Ammonium assimilation | [15]N spiking - incubation - mass-spectrometry | Rosette - Deck incubations | Tremblay J.E./ Raimbault P. | Y | 2, 3, 4 |
| [15]N-Ammonium assimilation | [15]N spiking - incubation - mass-spectrometry | Rosette In-situ production line | Tremblay J.E./ Raimbault P. | Y | 2, 3, 4 |
| [15]N-Ammonium oxidation (Nitrification) | [15]N spiking - incubation - mass-spectrometry | Rosette - Deck incubations | Tremblay J.E./ Raimbault P. | Y | 2, 3, 4 |
| [15]N-Ammonium oxidation (Nitrification) | [15]N spiking - incubation - mass-spectrometry | Rosette In-situ production line | Tremblay J.E./ Raimbault P. | Y | 2, 3, 4 |
| [15]N-Ammonium primary production ([13]C) | [15]N spiking - incubation - mass-spectrometry | Rosette - Deck incubations | Tremblay J.E./ Raimbault P. | Y | 2, 3, 4 |
| [15]N-Ammonium regeneration | [15]N spiking - incubation - mass-spectrometry | Rosette - Deck incubations | Tremblay J.E./ Raimbault P. | Y | 2, 3, 4 |
| [15]N-Ammonium regeneration | [15]N spiking - incubation - mass-spectrometry | Rosette In-situ production line | Tremblay J.E./ Raimbault P. | Y | 2, 3, 4 |
| [15]N-$N_2$ fixation | [15]N spiking - incubation - mass-spectrometry | Rosette water sample | Tremblay J.E./ Raimbault P. | N | 2, 3, 4 |
| [15]N-Nitrate assimilation | [15]N spiking - incubation - mass-spectrometry | Rosette - Deck incubations | Tremblay J.E./ Raimbault P. | Y | 2, 3, 4 |
| [15]N-Nitrate assimilation | [15]N spiking - incubation - mass-spectrometry | Rosette In-situ production line | Tremblay J.E./ Raimbault P. | Y | 2, 3, 4 |
| [210]Pb geochronology of core samples | [209]Po alpha spectrometry | Box corer | Rochon A. | N | 1 |
| [210]Pb geochronology of core samples | [209]Po alpha spectrometry | CASQ corer | Rochon A. | N | 1 |
| [234]Th (1 micron < particles > 70 micron) | Beta-counting | Foredeck In-situ pump | Gasser B. | Y | |
| [234]Th (particles > 70 micron) | Beta-counting | Foredeck In-situ pump | Gasser B. | Y | |
| [234]Th (Particulate) | Beta-counting | Drifting Sediment trap | Gasser B. | Y | 5 |
| AAPB (abundance) | IR microscopy, fluorimetry. FISH | Rosette water sample | Jeanthon C./ Boeuf D. | Y | 6 |
| AAPB (abundance) | IR microscopy, fluorimetry. FISH | Zodiac water sample | Jeanthon C./ Boeuf D. | Y | 6 |
| Absorption (particulate) | PSICAM | Barge water sample | Leymarie E. | Y | |
| Absorption (particulate) | PSICAM | Rosette water sample | Leymarie E. | Y | |
| Absorption (particulate) | Spectrophotometer (filters) | Barge water sample | Belanger S. | Y | 7, 8 |
| Absorption (particulate) | Spectrophotometer (filters) | Continuous on way | Belanger S. | Y | 7, 8 |
| Absorption (particulate) | Spectrophotometer (filters) | Rosette water sample | Belanger S. | Y | 7, 8 |
| Absorption (particulate) | Spectrophotometer (filters) | Zodiac profiler | Belanger S. | Y | 7, 8 |
| Absorption (total) | PSICAM | Barge water sample | Leymarie E. | Y | |
| Absorption (total) | PSICAM | Rosette water sample | Leymarie E. | Y | |
| Absorption coefficient (total) (9 wavelengths) | Wetlabs AC9 Serial# 156 | Rosette profiler | Ehn J. | Y | |
| Absorption coefficient (total) (9 wavelengths in IR | Wetlabs AC9 Serial# 303 | Barge profiler | Doxaran D. | Y | 9 |
| Absorption coefficient (total) (9 wavelengths) | Wetlabs AC9 Serial# 279 | Barge profiler | Doxaran D. | Y | 10, 11 |
| Air Relative Humidity | Humidity Sensor | Foredeck Meteorological Tower | Papakyriakou T. | Y | |
| Alkalinity total (TA) | Potentiometry | Barge water sample | Mucci A./ Lansard B. | Y | 12, 13, 14 |
| Alkalinity total (TA) | Potentiometry | Rosette | Mucci A./ Lansard B. | Y | 12, 13, 14 |
| Alkalinity total (TA) | Potentiometry | Zodiac water sample | Mucci A./ Lansard B. | Y | 12, 13, 14 |
| Alkanes | GC-MS | Box corer | Bouloubassi I. | Y | |
| Alkanes | GC-MS | CASQ corer | Bouloubassi I. | Y | |
| Ammonium ($NH_4^+$) photo-production apparent quantum yield (AQY) | sun simulator - fluorimetry | Rosette water sample | Xie H./ Tremblay J.E. | Y | 15 |
| Ammonium ($NH_4^+$) photo-production apparent quantum yield (AQY) | sun simulator - fluorimetry | Zodiac water sample | Xie H./ Tremblay J.E. | Y | 15 |
| Aragonite : saturation state | Derived parameter | Barge water sample | Mucci A./ Lansard B. | Y | 16 |
| Aragonite : saturation state | Derived parameter | Rosette water sample | Mucci A./ Lansard B. | Y | 16 |
| Aragonite : saturation state | Derived parameter | Zodiac water sample | Mucci A./ Lansard B. | Y | 16 |
| Attenuation coefficient (total) (9 wavelengths in IR) | Wetlabs AC9 Serial #0303 | Barge profiler | Doxaran D. | Y | 9 |
| Attenuation coefficient (total) (9 wavelengths) | Wetlabs AC9 Serial #156 | Rosette profiler | Ehn J. | Y | |
| Attenuation coefficient (total) (9 wavelengths) | Wetlabs AC9 Serial #279 | Barge profiler | Doxaran D. | Y | 10, 11 |
| Attenuation coefficient at 660 nm | Wetlabs (CRover) transmissometer | Drifting profiling float | Doxaran D. | Y | 17, 18 |
| Backscattering 532 nm | Wetlabs (ECO$^3$) backscatterometer | Drifting profiling float | Doxaran D. | Y | 17, 18 |
| Backscattering coefficient (3 wavelengths in IR) | Wetlabs ECO-BB3 serial #538 | Barge profiler | Doxaran D. | Y | 19, 20 |
| Backscattering coefficient (3 wavelengths) | Wetlabs ECO-BB3 serial #028 | Barge profiler | Doxaran D. | Y | 19, 20 |
| Backscattering coefficient (8 wavelengths, spectral) | Hydroscat-6 (ser#97074) and two a-Beta (HOBI-Labs) | Barge profiler | Reynolds R. | Y | 21, 22 |
| Backscattering coefficient (8 wavelengths, spectral) | Hydroscat-6 (ser#97074) and two a-Beta (HOBI-Labs) | Foredeck | Reynolds R. | Y | 21, 22 |
| Backscattering coefficient (9 wavelengths) | Wetlabs ECO-BB9 serial# 274 | Rosette profiler | Ehn J. | Y | |
| Bacteria (abundance) | Flow cytometry | Rosette water sample | Vaulot D. | Y | 23 |
| Bacterial bio-volume | Epifluorescence microscopy | Rosette water sample | Joux F./ Ortega-Retuerta E. | N | 24, 25 |
| Bacterial diversity | CE-SSCP and 454 Tag-Pyrosequencing | Rosette and Zodiac water sample | Joux F./J.F. Ghiglione | N | 26, 27 |

Table 1: Parameters measured during the MALINA oceanographic expedition. Parameters are ordered alphabetically. *(continued)*

| Parameters | Method | Sampling | Principal investigators | Included in the data repository | Reference |
|---|---|---|---|---|---|
| Bacterial growth (limitation by nutrients) | Leucine-[3]H incubations - cells counts | Rosette water sample | Joux F./ Jeffrey W./ Ortega-Retuerta E. | N | 25, 28 |
| Bacterial production | Leucine-[3]H incorporation | Rosette water sample | Joux F./ Jeffrey W. | Y | 25, 29, 30 |
| Bacterial production | Leucine-[3]H incorporation | Zodiac water sample | Joux F./ Jeffrey W. | Y | 25, 29, 30 |
| Bacterial production (effects of DOM UV exposure on...) | Leucine-[3]H incorporation - cell counts | Rosette water sample | Joux F./ Jeffrey W./ Ortega-Retuerta E. | N | 25 |
| Bacterial respiration (whole community) | $O^2$ consumption - Winkler - Incubations | Rosette water sample | Joux F./ Ortega-Retuerta E. | Y | 25, 31 |
| Benthic ammonium flux | Incubations - Colorimetry | Box corer | Link H./ Archambault P./ Chaillou G. | Y | 32, 33 |
| Benthic Macrofauna abundance | Microscopy | Box corer | Link H./ Archambault P./ Chaillou G. | N | 33, 34, 35 |
| Benthic Macrofauna diversity | Microscopy | Box corer | Link H./ Archambault P./ Chaillou G. | N | 33, 34, 35 |
| Benthic nitrate flux | Incubations - Colorimetry- Autoanalyzer | Box corer | Link H./ Archambault P./ Chaillou G. | Y | 32, 33, 34 |
| Benthic nitrite flux | Incubations - Colorimetry- Autoanalyzer | Box corer | Link H./ Archambault P./ Chaillou G. | Y | 32, 33, 34 |
| Benthic phosphate flux | Incubations - Colorimetry- Autoanalyzer | Box corer | Link H./ Archambault P./ Chaillou G. | Y | 32, 33, 34 |
| Benthic respiration | Incubations - Optic - Oxygen probe | Box corer | Link H./ Archambault P./ Chaillou G. | Y | 32, 33, 34 |
| Benthic silicic acid flux | Incubations - Colorimetry - Autoanalyzer | Box corer | Link H./ Archambault P./ Chaillou G. | Y | 32, 33, 34 |
| Calcite : saturation state | derived parameter | Rosette water sample | Mucci A./ Lansard B. | Y | 16 |
| Calcite : saturation state | Derived parameter | Barge water sample | Mucci A./ Lansard B. | Y | 16 |
| Calcite : saturation state | Derived parameter | Zodiac water sample | Mucci A./ Lansard B. | Y | 16 |
| Campesterol, cholesterol, sistosterol and products of degrad | GC-MS | Rosette water sample | Sempere R. | Y | 36, 37, 38 |
| CDOM absorption | PSICAM | Barge water sample | Leymarie E. | Y | |
| CDOM absorption | PSICAM | Rosette water sample | Leymarie E. | Y | |
| CDOM absorption | PSICAM | Zodiac water sample | Leymarie E. | Y | |
| CDOM absorption | Spectrophotometer | Barge water sample | Matsuoka A./ Bricaud A. | N | 41, 42 |
| CDOM absorption | Spectrophotometer | Rosette water sample | Matsuoka A./ Bricaud A. | N | 41, 42 |
| CDOM absorption | Spectrophotometer | Zodiac water sample | Matsuoka A./ Bricaud A. | N | 41, 42 |
| CDOM absorption | Ultrapath | Barge water sample | Bricaud A. | Y | 39, 40 |
| CDOM absorption | Ultrapath | Rosette water sample | Bricaud A. | Y | 39, 40 |
| CDOM absorption | Ultrapath | Zodiac water sample | Bricaud A. | Y | 39, 40 |
| CDOM fluorescence | Haardt fluorometer | Rosette profiler | Benner R./ Belanger S./ Amon/ Sempere R. | Y | 43, 44 |
| CDOM fluorescence | Wetlabs (ECO[3]) fluorometer | Drifting profiling float | Doxaran D. | Y | 18 |
| CDOM fluorescence | Wetlabs WetStar WSCD | Barge profiler | Doxaran D. | Y | 18 |
| CDOM fluorescence EEM (excitation-emission-matrix) | Spectrofluorimetry | Rosette water sample | Sempere R. | N | 45 |
| CDOM fluorescence EEM (excitation-emission-matrix) | Spectrofluorimetry | Zodiac water sample | Sempere R. | N | 45 |
| Chlorophyll a and Phaeopigments (concentration) | Fluorimetry Size fractionned | Rosette water sample | Gosselin M./ Belanger S. | Y | 46 |
| Chlorophyll a and Phaeopigments (benthic) | Fluorometric analysis | Box corer | Link H./ Archambault P./ Chaillou G. | Y | 32, 33, 34 |
| Chlorophyll a fluorescence [Fchla (z)] | Chelsea Mini-Track a II fluorometer | Barge profiler | Doxaran D. | Y | 18 |
| Chlorophyll a fluorescence [Fchla (z)] | SeaPoint fluorometer | Rosette profiler | Gratton Y./ Prieur L./ Tremblay J.E. | Y | |
| Chlorophyll a fluorescence [Fchla (z)] | Wetlabs (ECO[3]) fluorometer | Drifting profiling float | Doxaran D. | Y | 18 |
| CO photo-prod. apparent quantum yield for CDOM | Sun simulator - reduction gas analyzer | Rosette water sample | Xie H. | Y | 47 |
| CO photo-prod. apparent quantum yield for CDOM | Sun simulator - reduction gas analyzer | Zodiac water sample | Xie H. | Y | 47 |
| CO photo-prod. apparent quantum yield for particulate matter | Sun simulator - reduction gas analyzer | Rosette water sample | Xie H. | Y | 47 |
| CO photo-prod. apparent quantum yield for particulate matter | Sun simulator - reduction gas analyzer | Zodiac water sample | Xie H. | Y | 47 |
| $CO^2$ (atm) concentration | Infra Red | Foredeck Meteorological Tower | Papakyriakou T. | Y | |
| $CO^3$2- concentration | Derived parameter | Barge water sample | Mucci A./ Lansard B. | Y | 16 |
| $CO^3$2- concentration | Derived parameter | Rosette water sample | Mucci A./ Lansard B. | Y | 16 |
| $CO^3$2- concentration | Derived parameter | Zodiac water sample | Mucci A./ Lansard B. | Y | 16 |
| Coccolithophorids | Microscopy | Rosette water sample | Coupel P. | Y | |
| Conductivity (z) | Sensor on SBE Fascat CTD serial # | Barge profiler | Doxaran D. | Y | 48 |
| Conductivity (z) | Sensor SeaBird 4c on CTD SBE-911 | Rosette profiler | Gratton Y./ Prieur L. | Y | |
| CTD | Seabird | Drifting profiling float | Doxaran D. | Y | 48 |
| Cultures of sorted populations | Sorted by flow cytometry, serial dilution and single cell pipetting | Rosette water sample | Vaulot D. | N | 49 |
| delta [13]C - DIC | Mass Spectrometry (IRMS) | Barge water sample | Mucci A./ Lansard B. | Y | |
| delta [13]C - DIC | Mass Spectrometry (IRMS) | Rosette water sample | Mucci A./ Lansard B. | Y | |
| delta [13]C - DIC | Mass Spectrometry (IRMS) | Zodiac water sample | Mucci A./ Lansard B. | Y | |
| delta [13]C on suspended particulate matter | Mass Spectrometry | Rosette water sample | Tremblay J.E./ Raimbault P. | N | 50 |
| delta [18]O - water | Mass Spectrometry (IRMS) | Barge water sample | Mucci A./ Lansard B. | Y | 13 |
| delta [18]O - water | Mass Spectrometry (IRMS) | Rosette water sample | Mucci A./ Lansard B. | Y | 13 |
| delta [18]O - water | Mass Spectrometry (IRMS) | Zodiac water sample | Mucci A./ Lansard B. | Y | 13 |
| Diacids composition | GC/MS | Rosette water sample | Sempere R. | N | 51 |

Table 1: Parameters measured during the MALINA oceanographic expedition. Parameters are ordered alphabetically. *(continued)*

| Parameters | Method | Sampling | Principal investigators | Included in the data repository | Reference |
|---|---|---|---|---|---|
| Diacids composition | GC/MS | Zodiac water sample | Sempere R. | N | 51 |
| Diacids photo-production apparent quantum yield (AQY) | Sun simulator - GC/MS | Zodiac water sample | Sempere R. | N | 52 |
| Dinocyst, Pollen and Spores Abundance | Microscopy | Box corer | Rochon A. | N | 1 |
| Dinocyst, Pollen and Spores Abundance | Microscopy | CASQ corer | Rochon A. | N | 1 |
| Dinocyst, Pollen and Spores Identification | Microscopy | Box corer | Rochon A. | N | 1 |
| Dinocyst, Pollen and Spores Identification | Microscopy | CASQ corer | Rochon A. | N | 1 |
| Dinoflagellates cysts Abundance | Microscopy | Box corer | Rochon A. | N | 1 |
| Dinoflagellates cysts Abundance | Microscopy | CASQ corer | Rochon A. | N | 1 |
| Dinoflagellates cysts Identification | Microscopy | Box corer | Rochon A. | N | 1 |
| Dinoflagellates cysts Identification | Microscopy | CASQ corer | Rochon A. | N | 1 |
| Dissolved Inorg. Carbon photo-prod. apparent quantum yield | Sun simulator - indrared $CO_2$ analyzer | Rosette water sample | Xie H./ Belanger S. | Y | 53 |
| Dissolved Inorg. Carbon photo-prod. apparent quantum yield | Sun simulator - indrared $CO_2$ analyzer | Zodiac water sample | Xie H./ Belanger S. | Y | 53 |
| Dissolved Nitrogen (Total) (TDN) | High Temperature Catalytic Oxidation | Rosette water sample | Benner R. | Y | 54 |
| Dissolved Nitrogen (Total) (TDN) | High Temperature Catalytic Oxidation | Zodiac water sample | Benner R. | Y | 54 |
| Dissolved Organic Carbon (DOC) | High Temperature Catalytic Oxidation | Rosette water sample | Benner R. | Y | 54 |
| Dissolved Organic Carbon (DOC) | High Temperature Catalytic Oxidation | Rosette water sample | Sempere R. | Y | 45, 55 |
| Dissolved Organic Carbon (DOC) | High Temperature Catalytic Oxidation | Zodiac water sample | Benner R. | Y | 54 |
| Dissolved Organic Carbon (DOC) | High Temperature Catalytic Oxidation | Zodiac water sample | Sempere R. | Y | 45, 55 |
| Dissolved Organic Carbon (DOC) | Wet oxidation | Rosette water sample | Tremblay J.E./ Raimbault P. | Y | 56 |
| Dissolved Organic Nitrogen (DON) | Wet oxidation | Rosette water sample | Tremblay J.E./ Raimbault P. | Y | 56 |
| Dissolved Organic Phosphorus (DOP) | Wet oxidation | Rosette water sample | Tremblay J.E./ Raimbault P. | Y | 56 |
| Ed, Lu, Eu, Es | C-OPS package (320, 340, 380, 395 nm) | Barge profiler | Hooker | Y | 57 |
| Electric resistivity (sediment core physical properties) | Geotek Multi Sensor Core Logger | Box corer | Rochon A. | N | 1 |
| Electric resistivity (sediment core physical properties) | Geotek Multi Sensor Core Logger | CASQ corer | Rochon A. | N | 1 |
| Eukaryotes (abundance) | DAPI epifluorescence microscopy | Rosette water sample | Lovejoy C. | N | 58 |
| Eukaryotes (biomass) | DAPI epifluorescence microscopy | Rosette water sample | Lovejoy C. | N | 58 |
| $fCO_2$ | Derived parameter | Barge water sample | Mucci A./ Lansard B. | Y | 12 |
| $fCO_2$ | Derived parameter | Rosette water sample | Mucci A./ Lansard B. | Y | 12 |
| $fCO_2$ | Derived parameter | Zodiac water sample | Mucci A./ Lansard B. | Y | 12 |
| Foraminifera abundance | Microscopy | Box corer | Rochon A. | N | 1 |
| Foraminifera abundance | Microscopy | CASQ corer | Rochon A. | N | 1 |
| Foraminifera identification | Microscopy | Box corer | Rochon A. | N | 1 |
| Foraminifera identification | Microscopy | CASQ corer | Rochon A. | N | 1 |
| Gamma density (sediment core physical properties) | Geotek Multi Sensor Core Logger | Box corer | Rochon A. | N | 1 |
| Gamma density (sediment core physical properties) | Geotek Multi Sensor Core Logger | CASQ corer | Rochon A. | N | 1 |
| $H_2O$ (atm) concentration | Infrared gas analyzer | Foredeck Meteorological Tower | Papakyriakou T. | Y | |
| $HCO_3$- concentration | Derived parameter | Barge water sample | Mucci A./ Lansard B. | Y | 12, 16 |
| $HCO_3$- concentration | Derived parameter | Rosette water sample | Mucci A./ Lansard B. | Y | 12, 16 |
| $HCO_3$- concentration | Derived parameter | Zodiac water sample | Mucci A./ Lansard B. | Y | 12, 16 |
| Hydro SCAMP (Temp, Salin, Chlorophyll, turb. ...) | SCAMP profiler | In-water profiler | Gratton Y. | Y | |
| Hydrolysable Amino Acids (Total) (THAA) | HPLC | Rosette water sample | Benner R. | Y | 54 |
| Hydrolysable Amino Acids (Total) (THAA) | HPLC | Zodiac water sample | Benner R. | Y | 54 |
| Hydroxyl radicals (OH) | HPLC | Rosette water sample | Sempere R. | Y | |
| Hydroxyl radicals (OH) | HPLC | Zodiac water sample | Sempere R. | Y | |
| IP25 (C25 Monounsaturated Hydrocarbon) | GC | Box corer | Masse G. | Y | |
| IP25 (C25 Monounsaturated Hydrocarbon) | GC | CASQ corer | Masse G. | Y | |
| Irradiance | Satlantic (PUV) (305,325, 340, 380,..) | Foredeck | Sempere R. | Y | 45 |
| Irradiance (412, 490, 555 nm) | Satlantic (OCR) radiometer | Drifting profiling float | Doxaran D. | Y | 18 |
| Lignin phenols (dissolved) | GC/MS | Rosette water sample | Benner R. | Y | 59 |
| Lignin phenols (dissolved) | GC/MS | Zodiac water sample | Benner R. | N | 59 |
| Lipid biomarqueurs | GC-Flamme Ionization Detection / GC-MS | Box corer | Tolosa I. | Y | 60, 61 |
| Lipid biomarqueurs | GC-Flamme Ionization Detection / GC-MS | Foredeck in-situ pump | Tolosa I. | Y | 60, 61 |
| Lipid biomarqueurs d$^{13}$C | GC-Combustion Isotope ratio MS | Box corer | Tolosa I. | Y | 60, 61 |
| Lipid biomarqueurs d$^{13}$C | GC-Combustion Isotope ratio MS | Foredeck in-situ pump | Tolosa I. | Y | 60, 61 |
| Long-Wave radiation (Lwin) | Pyrgeometer | Wheel-house radiation platform | Papakyriakou T. | Y | |
| Magnetic susceptibility (sediment core physical properties) | Geotek Multi Sensor Core Logger | Box corer | Rochon A. | N | 1 |
| Magnetic susceptibility (sediment core physical properties) | Geotek Multi Sensor Core Logger | CASQ corer | Rochon A. | N | 1 |

| Parameters | Method | Sampling | Principal investigators | Included in the data repository | Reference |
|---|---|---|---|---|---|
| Major and minor elements | XRF core scanner | CASQ corer | Martinez P. | Y | |
| Nanoeukaryotes (abundance) | Flow cytometry | Rosette water sample | Vaulot D. | Y | 62 |
| $NH_4^+$ | Fluorescence | Rosette water sample | Tremblay J.E./ Raimbault P. | Y | 63 |
| Nitrate (concentration) | Satlantic ISUS | Rosette profiler | Gratton Y./ Prieur L./ Tremblay J.E. | Y | |
| $NO_2^-$ | Colorimetry/Autoanalyzer | Rosette water sample | Tremblay J.E./ Raimbault P. | Y | 64 |
| $NO_3^-$ | Colorimetry/Autoanalyzer | Rosette water sample | Tremblay J.E./ Raimbault P. | Y | 64 |
| Oxygen (dissolved) | Discrete samples Winkler Method | Rosette water sample | Prieur L. | Y | |
| Oxygen (dissolved) | Idronaut Ocean Seven $O^2$ sensor | Continuous horizontal | Papakyriakou T. | Y | |
| Oxygen (dissolved) | SeaBird SBE-43 sensor | Rosette profiler | Gratton Y./ Prieur L. | Y | |
| P-waves speed (sediment core physical properties) | Geotek Multi Sensor Core Logger | Box corer | Rochon A. | N | 1 |
| P-waves speed (sediment core physical properties) | Geotek Multi Sensor Core Logger | CASQ corer | Rochon A. | N | 1 |
| Paleomagnetism | Cryogenic magnetometer | Box corer | Rochon A. | N | 1 |
| Paleomagnetism | Cryogenic magnetometer | CASQ corer | Rochon A. | N | 1 |
| PAR | Biospherical sensor | Barge profiler | Wright V./ Hooker S. | N | 57 |
| PAR | Biospherical sensor | Rosette profiler | Gratton Y./ Prieur L./ Tremblay J.E. | Y | |
| PAR | PARLite sensor | Wheel-house radiation platform | Papakyriakou T. | Y | |
| Particle Size Distribution | Coulter counter | Barge water sample | Reynolds R. | Y | 21, 67 |
| Particle Size Distribution | LISST-100X | Barge profiler | Reynolds R. | Y | 21, 66 |
| Particle Size Distribution | LISST-100X | Rosette profiler | Reynolds R. | Y | 21, 66 |
| Particle Size Distribution | UVP-5 | In-water profiler | Picheral M. | Y | 65 |
| Particle Size Distribution | Coulter counter | Rosette water sample | Reynolds R. | Y | 21, 67 |
| Particulate Organic Carbon (POC) | CHN analyzer on SPM filters | Barge water sample | Doxaran D./ Ehn J./ Babin M. | Y | 68 |
| Particulate Organic Carbon (POC) | CHN analyzer on SPM filters | Rosette water sample | Doxaran D./ Ehn J./ Babin M. | Y | 68 |
| Particulate Organic Carbon (POC) | CHN analyzer on SPM filters | Zodiac water sample | Doxaran D./ Ehn J./ Babin M. | Y | 68 |
| Particulate Organic Carbon (POC) | Wet oxidation | Rosette water sample | Tremblay J.E./ Raimbault P. | Y | 69 |
| Particulate Organic Nitrogen (PON) | Wet oxidation | Rosette water sample | Tremblay J.E./ Raimbault P. | Y | 69 |
| Particulate Organic Phosphorus (POP) | Wet oxidation | Rosette water sample | Tremblay J.E./ Raimbault P. | Y | 69 |
| pH (NBS scale) | SeaBird SBE-18 sensor | Rosette profiler | Gratton Y./ Prieur L./ Tremblay J.E. | Y | |
| pH (total proton scale) | Derived parameter | Barge water sample | Mucci A./ Lansard B. | Y | 12, 16 |
| pH (total proton scale) | Derived parameter | Rosette water sample | Mucci A./ Lansard B. | Y | 12, 16 |
| pH (total proton scale) | Derived parameter | Zodiac water sample | Mucci A./ Lansard B. | Y | 12, 16 |
| pH (total proton scale) | Spectrophometry | Barge water sample | Mucci A./ Lansard B. | Y | 12, 16 |
| pH (total proton scale) | Spectrophotometry | Rosette water sample | Mucci A./ Lansard B. | Y | 12, 16 |
| pH (total proton scale) | Spectrophotometry | Zodiac water sample | Mucci A./ Lansard B. | Y | 12, 16 |
| Photoheterotrophs (DNA diversity) | DNA clone library | Rosette water sample | Jeanthon C./ Boeuf D. | Y | 6, 70 |
| Photosynthetic eukaryotes (diversity) | DNA clone library and TRFLP of sorted populations | Rosette water sample | Vaulot D. | N | 71 |
| Photosynthetic eukaryotes (diversity) | DNA from filters | Rosette water sample | Vaulot D. | N | 71 |
| Photosynthetic eukaryotes (morphology) | Scanning Electron Microscopy | Rosette water sample | Vaulot D. | N | 72 |
| Photosynthetic parameters | $^{14}$C incubations | Rosette water sample | Huot Y. | Y | 73 |
| Phytoplankton (abundance) | Inverted microscope | Rosette water sample | Gosselin M./ Belanger S. | Y | 46, 74 |
| Phytoplankton (taxonomy) | Inverted microscope | Rosette water sample | Gosselin M./ Belanger S. | Y | 46, 74 |
| Phytoplankton pigments | HPLC | Barge water sample | Wright V./ Hooker S. | Y | |
| Phytoplankton pigments | HPLC | Rosette water sample | Ras J./ Claustre H. | Y | |
| Picoeukaryotes (abundance) | Flow cytometry | Rosette water sample | Vaulot D. | Y | 62 |
| Picoplankton (diversity) | DNA amplicon library | Rosette water sample | Lovejoy C. | N | 75 |
| Picoplankton (diversity) | RNA amplicon library | Rosette water sample | Lovejoy C. | N | 75 |
| Plankton taxonomy | UVP-5 | In-water profiler | Picheral M./ Marec C. | Y | |
| PR-containing bacteria (abundance) | Q-PCR | Rosette water sample | Jeanthon C./ Boeuf D. | Y | 70 |
| Pressure (Barometric) | Pressure Sensor | Foredeck Meteorological Tower | Papakyriakou T. | Y | |
| Radiance | Camera Luminance | Profile mode | Antoine D./ Leymarie E. | Y | 76 |
| Radiance | Camera Luminance | Surface mode | Antoine D./ Leymarie E. | Y | 76 |
| Radiance : Sub Product : average cosines | Camera Luminence | Profile mode | Antoine D./ Leymarie E. | Y | 76 |
| Radiance : Sub Product : average cosines | Camera Luminence | Surface mode | Antoine D./ Leymarie E. | Y | 76 |
| Radiance : Sub Product : irradiance (E) | Camera Luminence | Profile mode | Antoine D./ Leymarie E. | Y | 76 |
| Radiance : Sub Product : irradiance (E) | Camera Luminence | Surface mode | Antoine D./ Leymarie E. | Y | 76 |
| Radiance : Sub Product : Lnadir | Camera Luminence | Profile mode | Antoine D./ Leymarie E. | Y | 76 |

| Parameters | Method | Sampling | Principal investigators | Included in the data repository | Reference |
|---|---|---|---|---|---|
| Radiance : Sub Product : Lnadir | Camera Luminence | Surface mode | Antoine D./ Leymarie E. | Y | 76 |
| Radiance : Sub Product : Qnadir | Camera Luminence | Profile mode | Antoine D./ Leymarie E. | Y | 76 |
| Radiance : Sub Product : Qnadir | Camera Luminence | Surface mode | Antoine D./ Leymarie E. | Y | 76 |
| Radiance : Sub Product : scalar irradiance (Escal) | Camera Luminence | Profile mode | Antoine D./ Leymarie E. | Y | 76 |
| Radiance : Sub Product : scalar irradiance (Escal) | Camera Luminence | Surface mode | Antoine D./ Leymarie E. | Y | 76 |
| Radiance (surface leaving radiance) | BIOSORS | Foredeck | Hooker | Y | |
| Radiance (surface leaving radiance) | Satlantic HyperSAS | Foredeck | Belanger S. | N | 77 |
| Radiance (surface leaving radiance) | TriOS above water sensor | Foredeck | Doxaran D. | N | 78 |
| Salinity | Salinometer | Barge water sample | Gratton Y./ Prieur L. | Y | |
| Salinity | Salinometer | Rosette water sample | Gratton Y./ Prieur L. | Y | |
| Salinity [S (z)] | Derived parameter | Rosette profiler | Gratton Y./ Prieur L./ Tremblay J.E. | Y | |
| Salinity [S (z)] | Derived parameter from SBE Fastcat LOC IOP pack. | Barge profiler | Doxaran D. | Y | 48 |
| Short-Wave radiation (Swin) | Pyranometer | Wheel-house radiation platform | Papakyriakou T. | Y | |
| Si (OH)$_4$ | Colorimetry/Autoanalyzer | Rosette water sample | Tremblay J.E./ Raimbault P. | Y | 64 |
| Soluble reactive phosphate | Colorimetry/Autoanalyzer | Rosette water sample | Tremblay J.E./ Raimbault P. | N | 64 |
| SPM (Suspended Particulate Material) | dry weight (gravimetry) | Barge water sample | Doxaran D./ Ehn J./ Babin M. | Y | 68 |
| SPM (Suspended Particulate Material) | dry weight (gravimetry) | Rosette water sample | Doxaran D./ Ehn J./ Babin M. | Y | 68 |
| SPM (Suspended Particulate Material) | dry weight (gravimetry) | Zodiac water sample | Doxaran D./ Ehn J./ Babin M. | Y | 68 |
| Sugars | HPLC | Rosette water sample | Sempere R. | N | 79 |
| Sugars | HPLC | Zodiac water sample | Sempere R. | N | 79 |
| Synechococcus (abundance) | Flow cytometry | Rosette water sample | Vaulot D. | Y | 62 |
| Temperature (Air) | Temperature Sensor | Foredeck Meteorological Tower | Papakyriakou T. | Y | |
| Temperature (Surface Skin) | IR transducer | Foredeck Meteorological Tower | Papakyriakou T. | Y | |
| Temperature [T (z)] | Sensor SeaBird 3plus on CTD SBE-911 | Rosette profiler | Gratton Y./ Prieur L./ Tremblay J.E. | Y | |
| Temperature [T (z)] | Temp sensor on SBE Fastcat CTD serial # | Barge profiler | Doxaran D. | Y | 48 |
| Total Inorganic Carbon (TIC) | Derived parameter | Barge water sample | Mucci A./ Lansard B. | Y | |
| Total Inorganic Carbon (TIC) | Derived parameter | Rosette water sample | Mucci A./ Lansard B. | Y | |
| Total Inorganic Carbon (TIC) | Derived parameter | Zodiac water sample | Mucci A./ Lansard B. | Y | |
| Total Organic Carbon (TOC) | Wet oxidation | Rosette water sample | Tremblay J.E./ Raimbault P. | Y | 56 |
| Total Organic Nitrogen (TON) | Wet oxidation | Rosette water sample | Tremblay J.E./ Raimbault P. | Y | 56 |
| Total Organic Phosphorus (TOP) | Wet oxidation | Rosette water sample | Tremblay J.E./ Raimbault P. | Y | 56 |
| Trace metals | X-Ray fluorescence spectroscopy | CASQ corer | Martinez P. | Y | |
| Urea (concentration) | Spectrophotometry | Rosette water sample | Tremblay J.E./ Raimbault P. | N | 64 |
| Volume Scattering Function (VSF) | Benchtop use of POLVSM | Barge water sample | Chami M. | N | 80, 81 |
| Volume Scattering Function (VSF) | Benchtop use of POLVSM | Rosette water sample | Chami M. | N | 80, 81 |
| Volume Scattering Function (VSF) | Benchtop use of POLVSM | Zodiac water sample | Chami M. | N | 80, 81 |
| Wind direction | Vane | Foredeck Meteorological Tower | Papakyriakou T. | Y | |
| Wind speed | Anemometer | Foredeck Meteorological Tower | Papakyriakou T. | Y | |

(1) Durantou, L. et al. (2012) Biogeosciences [10.5194/bg-9-5391-2012]; (2) Raimbault, P. et al. (1999) J. Geophys. Res.: Oceans [10.1029/1998JC900004]; (3) Raimbault, P. et al. (2008) Biogeosciences [10.5194/bg-5-323-2008]; (4) Fernández I., C et al. (2007) Marine Ecology Progress Series [10.3354/meps337079]; (5) Miquel, J.-C. et al. (2015) Biogeosciences [10.5194/bg-12-5103-2015]; (6) Boeuf, D. et al. (2013) FEMS Microbiology Ecology [10.1111/1574-6941.12130]; (7) Bélanger, S. et al. (2013) Biogeosciences [10.5194/bg-10-4087-2013]; (8) Röttgers, R. et al. (2012) Applied Optics [10.1364/AO.51.001336]; (9) Doxaran, D. et al. (2007) Optics Express [10.1364/OE.15.012834]; (10) Pegau, W. S. et al. (1997) Applied Optics [10.1364/AO.36.006035]; (11) Sullivan, J.M. et al. (2006) Applied Optics [10.1364/AO.45.005294]; (12) Mucci, A. et al. (2010) Journal of Geophysical Research [10.1029/2009JC005330]; (13) Lansard, B. et al. (2012) J. Geophys. Res.: Oceans [10.1029/2011JC007299]; (14) Dickson, A.G. et al. (2007) PICES Special Publication; (15) Xie, H. et al. (2012) Biogeosciences [10.5194/bg-9-3047-2012]; (16) Beaupré-Laperrière, Al. et al. (2020) Biogeosciences [10.5194/bg-17-3923-2020]; (17) Poteau, A. et al. (2017) Geophysical Research Letters [10.1002/2017GL073949]; (18) Xing, X. et al. (2012) J. Geophys. Res.: Oceans [10.1029/2011JC007632]; (19) Doxaran, D. et al. (2016) Optics Express [10.1364/OE.24.003615]; (20) Boss, E. et al. (2001) Applied Optics [10.1364/AO.40.005503]; (21) Reynolds, R.A. et al. (2016) Limnology and Oceanography [10.1002/lno.10341]; (22) Neukermans, G. et al. (2016) Limnology and Oceanography [10.1002/lno.10316]; (23) Marie, D. et al. (1997) Appl. environ. microb. [10.1128/AEM.63.1.186-193.1997]; (24) Cottrell, M.T. et al. (2003) Limnology and Oceanography [10.4319/lo.2003.48.1.0168]; (25) Ortega-Retuerta, E. et al. (2012) Biogeosciences [10.5194/bg-9-3679-2012]; (26) Ortega-Retuerta, E. et al. (2013) Biogeosciences [10.5194/bg-10-2747-2013]; (27) Ghiglione, J. F. et al. (2008) Biogeosciences [10.5194/bg-5-1751-2008]; (28) Ortega-Retuerta, E. et al. (2012) Polar Biology [10.1007/s00300-011-1109-8]; (29) Kirchman, D.L. (2018) Handbook of Methods in Aquatic Microbial Ecology [10.1201/9780203752746]; (30) Smith, D.C. et al. (1992) Marine Microbial Food Webs; (31) Carignan, R. et al. (1998) Canadian Journal of Fisheries and Aquatic Sciences [10.1139/cjfas-55-5-1078]; (32) Link, H. et al. (2013) Biogeosciences [10.5194/bg-10-5911-2013]; (33) Link, H. et al. (2019) NA [10.1594/PANGAEA.908091]; (34) Link, H. et al. (2013) PLoS ONE [10.1371/journal.pone.0074077]; (35) Wei, C.-L. et al. (2020) Diversity and Distributions [10.1111/ddi.13013]; (36) Rontani, J.-F. et al. (2014) Organic Geochemistry [10.1016/j.orggeochem.2014.06.002]; (37) Rontani, J.-F. et al. (2012) Biogeosciences [10.5194/bg-9-4787-2012]; (38) Rontani, J.-F. et al. (2012) Biogeosciences [10.5194/bg-9-3513-2012]; (39) Bricaud, Annick et al. (2010) Journal of Geophysical Research [10.1029/2009JC005517]; (40) Matsuoka, A. et al. (2012) Biogeosciences [10.5194/bg-9-925-2012]; (41) Matsuoka, Atsushi et al. (2011) Journal of Geophysical Research [10.1029/2009JC005594]; (42) Matsuoka, A. et al. (2014) Biogeosciences [10.5194/bg-11-3131-2014]; (43) Amon, R.M.W. (2003) Journal of Geophysical Research [10.1029/2002JC001594]; (44) Cooper, L.W. et al. (2005) J. Geophys. Res.: Biogeosci. [10.1029/2005JG000031]; (45) Para, J. et al. (2013) Biogeosciences [10.5194/bg-10-2761-2013]; (46) Blais, M. et al. (2017) Limnology and Oceanography [10.1002/lno.10581]; (47) Song, G. et al. (2013) Biogeosciences [10.5194/bg-10-3731-2013]; (48) Sea-Bird Scientific. (2017) Sea-Bird Scientific; (49) Balzano, S. et al. (2012) Biogeosciences [10.5194/bg-9-4553-2012]; (50) Andrisoa, A. et al. (2019) Marine Chemistry [10.1016/j.marchem.2019.03.003]; (51) Sempéré, R. et al. (2019) Global Biogeochemical Cycles [10.1029/2018GB006165]; (52) Tedetti, M. et al. (2007) J. Photoch. Photobio. A [10.1016/j.jphotochem.2006.11.029]; (53) Bélanger, S. et al. (2006) Global Biogeochemical Cycles [10.1029/2006GB002708]; (54) Shen, Y. et al. (2012) Biogeosciences [10.5194/bg-9-4993-2012]; (55) Sohrin, R. et al. (2005) Journal of Geophysical Research [10.1029/2004JC002731]; (56) Raimbault, P. et al. (1999) Marine Chemistry [10.1016/S0304-4203(99)00038-9]; (57) Hooker, Stanford B. et al. (2020) Biogeosciences [10.5194/bg-17-475-2020]; (58) Vaqué, D. et al. (2008) Limnology and Oceanography [10.4319/lo.2008.53.6.2427]; (59) Fichot, C.G. et al. (2016) Frontiers in Marine Science [10.3389/fmars.2016.00007]; (60) Tolosa, I. et al. (2013) Biogeosciences [10.5194/bg-10-2061-2013]; (61) Tolosa, I. et al. (2004) Journal of Chromatography A [10.1016/j.chroma.2004.06.037]; (62) Marie, D. et al. (2014) Cytom. Part. A. [10.1002/cyto.a.22517]; (63) Holmes, R.M. et al. (1999) Canadian Journal of Fisheries and Aquatic Sciences [10.1139/f99-128]; (64) Aminot, A. et al. (2007) Méthodes d'analyse en milieu marin; (65) Picheral, M. et al. (2010) Limnol. Oceanogr.-Meth [10.4319/lom.2010.8.462]; (66) Reynolds, R. A. et al. (2010) Journal of Geophysical Research [10.1029/2009JC005930]; (67) Runyan, H. et al. (2020) J. Geophys. Res.: Oceans [10.1029/2020JC016218]; (68) Doxaran, D. et al. (2012) Biogeosciences [10.5194/bg-9-3213-2012]; (69) Raimbault, P. et al. (1999) Marine Ecology Progress Series [10.3354/meps180289]; (70) Boeuf, D. et al. (2016) Frontiers in Microbiology [10.3389/fmicb.2016.01584]; (71) Balzano, S. et al. (2012) ISME Journal [10.1038/ismej.2011.213]; (72) Balzano, S. et al. (2017) Journal of Phycology [10.1111/jpy.12489]; (73) Marcel, Babin et al. (1994) Limnology and Oceanography [10.4319/lo.1994.39.3.0694]; (74) Ardyna, M. et al. (2017) Limnology and Oceanography [10.1002/lno.10554]; (75) Comeau, A.M. et al. (2012) Scientific Reports [10.1038/srep00604]; (76) Antoine, D. et al. (2013) Journal of Atmospheric and Oceanic Technology [10.1175/JTECH-D-11-00215.1]; (77) Mobley, C.D. (1999) Applied Optics [10.1364/AO.38.007442]; (78) Ruddick, K.G. et al. (2006) Limnology and Oceanography [10.4319/lo.2006.51.2.1167]; (79) Panagiotopoulos, C. et al. (2014) Marine Chemistry [10.1016/j.marchem.2014.09.004]; (80) Chami, M. et al. (2014) Optics Express [10.1364/OE.22.026403]; (81) Harmel, T. et al. (2016) Optics Express [10.1364/OE.24.00A234]

*Author contributions.* Marcel Babin and Simon Bélanger designed the MALINA project, including the scientific objectives and the sampling strategy. Philippe Massicotte prepared the initial draft of the manuscript. Simon Bélanger was responsible for spectrophotometric measurement of particles absorption and fluorometric chlorophyll *a* determination onboard and to above-water radiometry (continuous incident irradiance measurements, etc). Jens Ehn was the logistic coordinator for the cruise and was responsible for IOPs measurements from the CTD-rosette and contributed to SPM, POC and particulate absorption measurements and processing. Pingqing Fu performed the GC/MS experiments on organic molecular compositions. Alfonso Mucci and Bruno Lansard were in charge of determining carbonate system parameters (pH and total alkalinity) as well as the stable oxygen isotopic composition of the water and stable carbon isotopic composition of the dissolved inorganic carbon. Bruno Lansard participated in the cruise, collected the samples as well as compiled and processed most of the data. Carbonate system parameters were used to determine the saturation of waters concerning aragonite as well as calculate the surface water $P_{CO2}$. These parameters and others were used by the proponents to identify and estimate the contribution of parental waters to the structure of the water column in the study area. Claudie Marec, Louis Prieur and Yves Gratton operated the rosette during the Malina Cruise. Claudie Marec was in charge of all the physical instruments during the cruise. She also operated the UVP5 sensor. Yves Gratton was in charge of the physical data processing during and after the cruise. Claudie Marec and Louis Prieur processed the LADCP data during the cruise. Marc Picheral developed, operated the UVP5 sensor and processed the data. Gabriel Gorsky contributed to the UVP development. Christian Jeanthon processed samples during the MALINA cruise, designed the experiments on photoheterotrophic bacteria and wrote the related manuscripts. Dominique Boeuf designed and performed the experiments on photoheterotrophic bacteria and wrote the related manuscripts. Daniel Vaulot and Dominique Marie participated in the Malina cruise and sampled for flow cytometry, DNA and cultures. Dominique Marie performed flow cytometry measurements and cell sorting. Daniel Vaulot also coordinated the work on phytoplankton culture and molecular analyses. Sergio Balzano performed molecular analyses of flow cytometry-sorted photosynthetic pico- and nano-eukaryotes as well as phytoplankton cultures. Priscilla Gourvil isolated and characterized phytoplankton cultures. Wade Jeffrey, Eva Ortega-Retuerta and Fabien Joux participated in the Malina cruise, collected samples for bacterial production and respiration and prokaryotic diversity, and performed nutrient limitation experiments. Ronald Benner participated in the MALINA cruise and collected the samples for DOC, TDN, THAA, and $TDLP_9$. Cédric Fichot analyzed the samples for DOC, TDN and $TDLP_9$. Yuan Shen analyzed the samples for THAA. Rainer Amon contributed advice for the leads and the measurements of CDOM fluorescence. Pascal Guillot was in charge of the processing and quality control of the CTD data. Connie Lovejoy, contributed material, protocols and advice for DNA sampling for planktonic material. Hermann J. Heipieper contributed to the data interpretation regarding bacterial diversity and activity. Huixiang Xie and Guisheng Song participated in the MALINA cruise, designed and conducted the onboard CO photoproduction experiment, and collected samples for the CO2 photoproduction experiment. Tim Papakyriakou oversaw the measurement of meteorological elements from an instrumented tower on the ship's foredeck, and the measurement of incoming solar shortwave, long-wave, PAR and UV radiation from sensors mounted on top of the ship's wheelhouse. Flavienne Bruyant and Yannick Huot participated in the MALINA cruise and performed the P vs E curves measurements, the FIRe fluorometer measurement (phytoplankton photosynthesis efficiency) and sampled for the photosynthetic protein analysis. Flavienne Bruyant also helped gather and merge the initial form of the MALINA database with the help of Claude-Anne Bouin who coded for SQL formatting and Marie-Hélène Forget who coordinated the effort. David Doxaran participated in the Malina field campaign, measured from the barge the water temperature, salinity and inherent optical properties as a function of depth, then processed the data. He was also in charge of measuring the suspended particulate matter and particulate organic carbon concentrations on collected water samples. Rick Reynolds and Guangming

Zheng participated in the MALINA field program and collected data on seawater optical properties and particle characteristics. Together with Dariusz Stramski, they conducted processing and curation of these data, formal analysis, and publication of re-
sults. All three authors have reviewed and edited the current manuscript for accuracy. David Antoine participated in MALINA Leg 1. He carried out radiometry measurements including from the underwater radiance cameras, and then processing, analysis and publication of the results. He has contributed to editing the current manuscript. Philippe Archambault contributed material, protocols and advice of benthic sampling and participated in the interpretation of benthic biodiversity and biodiversity-ecosystem functioning through his ArcticNet funding. Heike Link participated in the Malina cruise, was in charge of sediment sampling with
the box corer, designed and performed benthic incubation experiments onboard, analyzed sediment samples for pigments and macrofauna, processed, analyzed, interpreted and published results obtained from benthic incubations (benthic nutrient fluxes and respiration, macrofauna, TOC, pigments) and contributed to writing this manuscript. Edouard Leymarie participated in the Malina cruise and was involved in the radiance camera measurements and data processing as well as POLVSM instrument. Catherine Schmechtig gathered data and metadata for the Malina cruise through INSU-CNRS (Institut National des Sciences de
l'Univers - Centre National de la Recherche Scientifique) fundings. Malik Chami and Alexandre Thirouard participated in the MA-LINA cruise; they operated the POLVSM instrument to measure the particulate scattering phase function of the water sample. Beat Gasser and Jacobo Martín participated in the Malina cruise, performed sampling of suspended particles with foredeck in-situ pumps and assisted Claudie Marec in the deployment of the drifting sediment-trap line. Both performed sample treatment and analysis of $234_{Th}$ and Juan Carlos Miquel participated in the interpretation of the data and coordination of these operations
prior to the cruise. Imma Tolosa performed data analysis and interpretation of lipid biomarkers and their carbon isotopic ratios in suspended particulate matter and sediment, sampled respectively with the foredeck in-situ pumps and box corer. Vanessa McKague participated in the Malina cruise, mainly on the barge to collect inherent optical properties deploying the optical package to obtain vertical profiles. Nicole Garcia, Patrick Raimbault and Pierre Coupel have participated in the Malina cruise being in charge of collected samples for inorganic and organic nutrients as well as experimental incubations for the determination
of primary production and nitrogen assimilation rates. Nicole Garcia and Patrick Raimbault have performed laboratory nutrient analysis and isotopic measurements, respectively. Patrick Raimbault has compiled and processed the data and has contributed to editing the current manuscript. Jean-François Rontani carried out lipid analyzes (lipid extraction, derivatization and GC-MS analyzes) on SPM samples collected during the Malina cruise, interpreted the results obtained and participated in the writing of the present paper. Bruno Charrière participated in the Malina cruise, being in charge of sampling and analysis for DOC, CDOM,
monosaccharides, diacids and atmospheric-underwater PAR/UV radiation. Atsushi Matsuoka and Annick Bricaud participated in the Malina cruise, measured CDOM absorption spectra, processed and analyzed the data, and published the results. Richard Sempéré defined sampling strategy on DOC, CDOM, monosaccharides, diacids and atmospheric-underwater PAR/UV radiation, interpreted the results obtained and wrote the related manuscripts. Christos Panagiotopoulos performed monosaccharides analysis and wrote the related manuscript. Mickael Vaitilingom performed diacids analysis and wrote the related manuscript.
Serge Heussner interpreted the mass fluxes data and wrote the related paper. Michel Gosselin contributed material, equipment, protocols, and advice for the determination of chlorophyll *a* and phaeopigments in the water column (fluorometric method) and participated in the interpretation of the phytoplankton biomass and community composition data. Nicole Delsaut performed sediment trap samples preparation. André Rochon and Guillaume Massé designed the M.Sc. project to study the evolution of sea surface conditions in the Beaufort Sea and Lise Durantou carried out the analyses and interpreted the data. Stanford B.

Hooker co-developed the C-OPS instrumentation, designed the experiments and co-developed the software to acquire the data by launching a small vessel from the much larger icebreaker, and co-developed the software to process the data.

*Competing interests.*  The authors declare no competing interests.

*Acknowledgements.*  This work is dedicated to the memory of Captain Marc Thibault (commanding officer of the *CCGS Amundsen*, Canadian Coast Guard), Daniel Dubé (*CCGS Amundsen* helicopter pilot, Transport Canada) and Dr. Klaus Hochheim (research sci-
entist at the Centre for Earth Observation Science, University of Manitoba) who died in the *CCGS Amundsen* helicopter crash on the evening of 2013-09-09 in the icy waters of McClure Strait in the Canadian Arctic. We are very grateful to the captain (Marc Thibault) and crews of the Canadian research icebreaker *CCGS Amundsen* during the Malina cruise in the Beaufort Sea. This study was conducted as part of the Malina scientific program funded by ANR (Agence Nationale de la Recherche), INSU-CNRS (Institut National des Sciences de l'Univers - Centre National de la Recherche Scientifique), the LEFE-CYBER program, CNES (Centre
National d'Études Spatiales), the European Commission (Marie Curie program), ESA (European Space Agency) and ArcticNet, US NSF grants 0713915 and 1504137 to R.B. and 0229302, 0425582, 0713991 to R.M.W.A. The International Atomic Energy Agency is grateful to the Government of the Principality of Monaco for the support provided to its Environment Laboratories. Abderrahmane Nassim Taalba performed the $CO_2$ photoproduction experiment. ArcticNet and Québec-Océan funded several students and analyses for their project.

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
