# Peer review of "The Malina oceanographic expedition: How do changes in ice cover, permafrost and UV radiation impact biodiversity and biogeochemical fluxes in the Arctic Ocean?"

_Earth System Science Data, 2020_

## Referee Comment (RC1) · Anonymous Referee #1 · 26 Oct 2020

The MALINA bio-optical data set for the Arctic Ocean is a unique asset because it covers both open ocean and shallow, near shore waters around a large Arctic River, an area of high research interest right now. This paper presents a clear summary of the data collected, figures for some variables and a list of all variables measured and the corresponding scientist whose lab carried out the measurements. I downloaded the compiled data file from SEANOE and it is very easy to follow. I confirmed that the raw UVP5 large particulate data and images are available from the Ecotaxa website at Villefranche. I also confirmed that the code for the manuscript and figures is readily

available at the zenodo.org website. ## The article is itself appropriate to support the publication of the MALINA data set. However, I could not find the metadata for each variable listed in Table 1. The manuscript lists the current address of the person(s) shown on Table 1 for each variable in case of questions about the protocol used; however, after a few years, it is very probable that the address will no longer be valid. The text sometimes indicates the published manuscript that describes the sampling protocol used for that variable, but it is not done in a consistent manner for each variable. ## A few examples: the pH sensor was replaced by a CDOM fluorometer on the CTD [L75] - which paper describes that step and any calibration for both sensors? Just a few lines above, Guillot and Gratton 2010 are listed for the rosette data protocols; does this also apply to the CDOM data? [different from section 4.3.2]. Or the LISST-100X protocol (L165, no reference associated in the text). Which reference is the one for nitrification and ammonium regeneration [L315]? Not Ardyna 2017 nor the literature compared with the MALINA results (Ortega-Retuerte 2012, Le Fouest 2013) nor Tremblay 2014 [only inor and org nutrient concentrations and only rates for C and N uptake, not regeneration). Anyhow, this is but a handful of examples. I urge the authors to indicate throughout the text and/or in Table 1 the publication(s) and/or metadata where the respective sampling method details are described for the measurements of every single parameter listed in Table 1. In one place. Instead of having to find it several years later, if one is lucky. This would be a GREAT service to all future data users/chasers! ## With respect to the answers to the following questions: "Are error estimates and sources of errors given (and discussed in the article)? Are the accuracy, calibration, processing, etc. state of the art? Are common standards used for comparison?" These answers are NOT in the current text. They are likely in the MALINA manuscripts which are too many for this reviewer to cross check. I leave this decision to the Editor. The MALINA data set is incredibly significant for Arctic Ocean researchers (empiricists and especially modelers)– it is incredibly unique, of extremely high quality for a field-collected data set and very, very useful, and -I assume- complete. ## The manuscript is sufficient as written to explain the data set, List 1 is superb (though it will be complete with

the associated publication that describes the sampling or analytical protocol used) and the figures are good. Not sure all the figures are necessary; they are one fig showing the spatial distribution of several but not all the variables measured. ## Finally, "By reading the article and downloading the data set, would you be able to understand and (re-)use the data set in the future?" => Absolutely!!

---

## Short Comment (SC1) · 2 Nov 2020

Thank you for your comments and suggestions. Please find below our points-by-points answers.

Anonymous Referee #1 The MALINA bio-optical data set for the Arctic Ocean is a unique asset because it covers both open ocean and shallow, near shore waters around a large Arctic River, an area of high research interest right now. This paper presents a clear summary of the data collected, figures for some variables and a list

[Figure]

of all variables measured and the corresponding scientist whose lab carried out the measurements. I downloaded the compiled data file from SEANOE and it is very easy to follow. I confirmed that the raw UVP5 large particulate data and images are available from the Ecotaxa website at Villefranche. I also confirmed that the code for the manuscript and figures is readily available at the zenodo.org website.

Comment C1

The article is itself appropriate to support the publication of the MALINA data set. However, I could not find the metadata for each variable listed in Table 1. The manuscript lists the current address of the person(s) shown on Table 1 for each variable in case of questions about the protocol used; however, after a few years, it is very probable that the address will no longer be valid. The text sometimes indicates the published manuscript that describes the sampling protocol used for that variable, but it is not done in a consistent manner for each variable. A few examples: the pH sensor was replaced by a CDOM fluorometer on the CTD [L75] - which paper describes that step and any calibration for both sensors? Just a few lines above, Guillot and Gratton 2010 are listed for the rosette data protocols; does this also apply to the CDOM data? [different from section 4.3.2]. Or the LISST-100X protocol (L165, no reference associated in the text). Which reference is the one for nitrification and ammonium regeneration [L315]? Not Ardyna 2017 nor the literature compared with the MALINA results (Ortega-Retuerte 2012, Le Fouest 2013) nor Tremblay 2014 [only inor and org nutrient concentrations and only rates for C and N uptake, not regeneration). Anyhow, this is but a handful of examples. I urge the authors to indicate throughout the text and/or in Table 1 the publication(s) and/or metadata where the respective sampling method details are described for the measurements of every single parameter listed in Table 1. In one place. Instead of having to find it several years later, if one is lucky. This would be a GREAT service to all future data users/chasers!

Answer A1

The reviewer raises an important point here. It is to be noted that a sentence in section 6 (Code and data availability) is already pointing out toward the LEFE-CYBER website which contains all the metadata information:

The raw data provided by all the researchers, as well as meta-data, are available on the LEFE-CYBER repository (http://www.obs-vlfr.fr/proof/php/malina/x_datalist_1.php?xxop=malina&xxcamp=malina).

We however agree with the recommendation of providing more information on the metadata and the methodological details associated with each measured variable. Table 1 will be modified to add and refer the readers toward existing MALINA publications that provide in-depth methodological information for the measured variables.

Comment C2

With respect to the answers to the following questions: "Are error estimates and sources of errors given (and discussed in the article)? Are the accuracy, calibration, processing, etc. state of the art? Are common standards used for comparison?" These answers are NOT in the current text. They are likely in the MALINA manuscripts which are too many for this reviewer to cross check. I leave this decision to the Editor. The MALINA data set is incredibly significant for Arctic Ocean researchers (empiricists and especially modelers)– it is incredibly unique, of extremely high quality for a field-collected data set and very, very useful, and -I assume- complete.

Answer A2

We agree that providing detailed information about the accuracy and calibration processes are important. However, it would be cumbersome to provide such information for all the measured parameters. The readers will be invited to consult the additional list of references provided in answer A1 (Table 1) where each specific MALINA research paper is likely to provide such information.

Comment C3

The manuscript is sufficient as written to explain the data set, List 1 is superb (though it will be complete with the associated publication that describes the sampling or analytical protocol used) and the figures are good. Not sure all the figures are necessary; they are one fig showing the spatial distribution of several but not all the variables measured.

Answer A3

For the associated publications, this was addressed in answers A1 and A2. As for the figures, we decided to present a rapid overview of what we considered the basic and most important parameters in relation to the main objectives of the MALINA research project. We are happy to reconsider the choice of the figures if the editor or the reviewers have specific queries.

Comment C4

Finally, "By reading the article and downloading the data set, would you be able to understand and (re-)use the data set in the future?" => Absolutely!!

Answer A4

Thank you very much for this comment!

---

## Referee Comment (RC2) · Anonymous Referee #2 · 24 Jan 2021

General comment This paper provides a good description of an excellent program focused on an environmental system and its likely response to effects of climate change. The large sampling area of multiple along-shelf transects, and sections running from very shallow waters at the river mouth out to the deeper basin provide great coverage. The multi-disciplinary research makes these data particularly important in answering the direct study questions and for use by the wider science community. This is a unique and valuable data set. The publication is well structured and clear. The figures, in particular the map of station location, the Mackenzie River runoff, and water masses were

clear and helpful for understanding the data set and the rest of the paper. The data and code are accessible through the paper's given websites (LEFE-CYBER for raw and meta data, SEANOE for final data, Ecotaxa for UVP5 data, and Zenodo for code). My two main comments are that the size of the main data set and lack of clear variable definitions make it very difficult to use, and the other follows up on Referee1's comment regarding incomplete information on methods and errors. It is a herculean task to bring together over 150 variables from over 50 PIs into a cohesive data set. With more information regarding the methods and errors and reworking the data set this will be a great addition for arctic and climate change scientists.

Specific Comments Methods and errors: The collection and analysis methods are well described for many of the variables, are current state of the art and have had quality control applied. However, as mentioned by Referee 1, the methods and information on errors are not given for all variables. I found it quite helpful to look at the LEFE-CYBER website (http://www.obs-vlfr.fr/proof/php/malina/x_datalist_1.php?xxop=malina&xxcamp=malina) mentioned by the author as home for metadata and raw data, but see that they do not have information documents for all the variables. Looking through the data folders there are some documents that exist but have not been added as links to the site's variable list (ex. CTD documentation in basic files/doc and basic files/report), but in other cases appear missing, for example "Chlorophyll-a and Phaeopigments (concentration)". It sounds like this will be addressed as mentioned in author's answers A1 and A2 to Referee 1 and information will be added to table 1 as to where to find methods and error information.

Data availability: It would help to have Table 1., the heart of the paper with key information, expanded to state if the listed variables are present in the provided data set. For example, oxygen Winkler data are not present. I assume the list on the LEFE-CYBER site is current and shows the other missing data. Having a table of all the variables collected is very useful and a user can follow up by contacting the PI, however it would

be good to know what is actually provided in the paper's data set.

Dataset: The large amount and various types of data pose a challenge in assembling a data collection. I did not find the current form of the final data posted to the SEANOE website useable due to its unwieldly size, structure and unclear variable names. A. The full data set is given in a single excel file approximately 2400 columns by 600,000 rows. The data set is too large to handle for software and users. My computer could open the file in excel but did not have enough memory to sort the rows. Its not practical for a user to sort through 2400 columns to find the variables of interest. A lookup table would help. This file could be separated into smaller files, in particular those data sets with a substantial number of columns (ANAP, AP and APHY with 500 columns each), and those that are not directly related (i.e. CTD, Foredeck meteorogical data, Benthic Boxcore, Zooplankton sampling). Remove other cruise data. There are data from programs in Aug 2008, Feb 2009, and the TARA programs from Jan 2010 to Aug 2012. Remove columns associated with these other programs (i.e. CTD data starting with column CNZ are for the TARA programs)

B. Variable names are not clear enough to determine what they are, which instrument, method or PI they are from, and what their unit of measurement is (i.e. "Average","345_1_AC"). Some variable names are shared between some data sets ("Cast", "Station", "Temperature", "Bottle"), some are not ("Temperature v. "Temp_Celsius", "Trans" v. "Xmiss", "Bottle" v. "N_btl_fired" v. "Niskin", "Station v. Q_Name").

In addition to clarifying what the variables and units are, it is important to be able to understand which variables connect to the information provided in the paper and Table 1. This currently is not practically possible. This could be addressed in a lookup table or with the variable name and header.

C. The many analyses from a given Niskin bottle do not use the same identifiers making it difficult to match all the data associated with a given water sample. The appended data set use a mix between Cast, Station, Bottle (and its variants), and Depth. At

best these data would be pre-joined, but at least it would be good to have a consistent identifier.

D. The date variable (column T) is a mixture of formats, only some currently understood by excel and thus not currently sortable. Corrections needed w/in data set: The CTD bottle data is one of the key parts to this data set that all water samples are linked back to. This data needs variable name "Depth_dbar" changed to "Pressure_dbar". The CTD+IOP data from the barge have been merged without any identifier to the date, cast, station or location. From the info sheet on the LEFE-CYBER page it looks like two steps are needed – the filename includes the IOP cast # and the infosheet has a table converting IOP cast # to station.

Specific comments on article's content Sea-ice cover The paper mentions the sea ice in Line 49 and 50 (shelf was not ice-free until mid-August) and Line 99 (shelf was ice-free). These appear inconsistent and it would be good to harmonize these lines. Perhaps an image of sea-ice concentration could be added to the paper within the environmental conditions section or a sea-ice edge added to the map of stations. Station order Add a further comment regarding the effect of sampling out of order. Did wind events occur between station sampling that may affect results? Even a statement saying this was not an issue or may be an issue would be helpful for a data user.

Technical Comments/Corrections Figure 8. Add "Surface samples" to the caption for clarity; It would be easier to compare if both sections were on the same plot. There are a number of "?" in the text that appear to be waiting for more information Figure 11a caption"?" , Line 270 "?", Line 279 "?", Line 282 "?" Here suggest using the pre-defined (Line 252) SCM instead of DCM for consistency. Figure 14A Would be nice to see A and B use same latitude range Line 380 Looks like this is supposed to be Station 690, not 680 Line 489: Title is missing text. "A 50 % increase in the mass of terrestrial. . ."

---

## Author Response (AR1)

Thank you for your comments and suggestions. Please find below our points-by-points answers.

**Anonymous Referee #1**

*The MALINA bio-optical data set for the Arctic Ocean is a unique asset because it covers both open ocean and shallow, near shore waters around a large Arctic River, an area of high research interest right now. This paper presents a clear summary of the data collected, figures for some variables and a list of all variables measured and the corresponding scientist whose lab carried out the measurements. I downloaded the compiled data file from SEANOE and it is very easy to follow. I confirmed that the raw UVP5 large particulate data and images are available from the Ecotaxa website at Villefranche. I also confirmed that the code for the manuscript and figures is readily available at the zenodo.org website.*

**Comment C1**

*The article is itself appropriate to support the publication of the MALINA data set. However, I could not find the metadata for each variable listed in Table 1. The manuscript lists the current address of the person(s) shown on Table 1 for each variable in case of questions about the protocol used; however, after a few years, it is very probable that the address will no longer be valid. The text sometimes indicates the published manuscript that describes the sampling protocol used for that variable, but it is not done in a consistent manner for each variable. A few examples: the pH sensor was replaced by a CDOM fluorometer on the CTD [L75] - which paper describes that step and any calibration for both sensors? Just a few lines above, Guillot and Gratton 2010 are listed for the rosette data protocols; does this also apply to the CDOM data? [different from section 4.3.2]. Or the LISST-100X protocol (L165, no reference associated in the*

*text). Which reference is the one for nitrification and ammonium regeneration [L315]? Not Ardyna 2017 nor the literature compared with the MALINA results (Ortega-Retuerte 2012, Le Fouest 2013) nor Tremblay 2014 [only inor and org nutrient concentrations and only rates for C and N uptake, not regeneration). Anyhow, this is but a handful of examples. I urge the authors to indicate throughout the text and/or in Table 1 the publication(s) and/or metadata where the respective sampling method details are described for the measurements of every single parameter listed in Table 1. In one place. Instead of having to find it several years later, if one is lucky. This would be a GREAT service to all future data users/chasers!*

**Answer A1**

The reviewer raises an important point here. It is to be noted that a sentence in section 6 (Code and data availability) is already pointing out toward the LEFE-CYBER website which contains all the metadata information:

*The raw data provided by all the researchers, as well as metadata, are available on the LEFE-CYBER repository (http://www.obs-vlfr.fr/proof/php/malina/x_datalist_1.php?xxop=malina&xxcamp=malina).*

We however agree with the recommendation of providing more information on the metadata and the methodological details associated with each measured variable. A new column in table 1 now presents an exhaustive list of MALINA publications (approximately 80) that provide in-depth methodological information for the measured variables.

**Comment C2**

*With respect to the answers to the following questions: "Are error estimates and sources of errors given (and discussed in the article)? Are the accuracy, calibration, processing, etc. state of the art? Are common standards used for comparison?" These answers are*

*NOT in the current text. They are likely in the MALINA manuscripts which are too many for this reviewer to cross check. I leave this decision to the Editor. The MALINA data set is incredibly significant for Arctic Ocean researchers (empiricists and especially modelers)– it is incredibly unique, of extremely high quality for a field-collected data set and very, very useful, and -I assume- complete.*

**Answer A2**

We agree that providing detailed information about the accuracy and calibration processes are important. However, it would be cumbersome to provide such information for all the measured parameters. The readers will be invited to consult the additional list of references provided in answer A1 (Table 1) where each specific MALINA research paper is likely to provide such information.

**Comment C3**

*The manuscript is sufficient as written to explain the data set, List 1 is superb (though it will be complete with the associated publication that describes the sampling or analytical protocol used) and the figures are good. Not sure all the figures are necessary; they are one fig showing the spatial distribution of several but not all the variables measured.*

**Answer A3**

For the associated publications, this was addressed in answers A1 and A2. As for the figures, we decided to present a rapid overview of what we considered the basic and most important parameters in relation to the main objectives of the MALINA research project. We are happy to reconsider the choice of the figures if the editor or the reviewers have specific queries.

**Comment C4**

*Finally, "By reading the article and downloading the data set, would you be able to understand and (re-)use the data set in the future?" => Absolutely!!*

**Answer A4**

Thank you very much for this comment!

**Anonymous Referee #2**

*General comment This paper provides a good description of an excellent program focused on an environmental system and its likely response to effects of climate change. The large sampling area of multiple along-shelf transects, and sections running from very shallow waters at the river mouth out to the deeper basin provide great coverage. The multi-disciplinary research makes these data particularly important in answering the direct study questions and for use by the wider science community. This is a unique and valuable data set. The publication is well structured and clear. The figures, in particular the map of station location, the Mackenzie River runoff, and water masses were clear and helpful for understanding the data set and the rest of the paper. The data and code are accessible through the paper's given websites (LEFE-CYBER for raw and meta data, SEANOE for final data, Ecotaxa for UVP5 data, and Zenodo for code). My two main comments are that the size of the main data set and lack of clear variable definitions make it very difficult to use, and the other follows up on Referee1's comment regarding incomplete information on methods and errors. It is a herculean task to bring together over 150 variables from over 50 PIs into a cohesive data set. With more information regarding the methods and errors and reworking the data set this will be a great addition for arctic and climate change scientists.*

**General answer**

Thank you for your comments. In the following sections, we present all the changes that we have made to make the data more structured and thus easier to use. Instead of providing a single file containing all the merged data, we are now providing a collection of smaller files that regroup all the data of the same type. In addition, we also provide a lookup table file that contains information that can be used to merge the data files together. See also answer A7 where we provide

detailed information on the change applied to the data files. We also removed entries in table 1 for which no data was available. We further added a list of new references (~80) associated to the variables presented in table 1. These references can be used to get more information about methods and errors associated to each variable.

**Comment C5**

*Specific Comments Methods and errors: The collection and analysis methods are well described for many of the variables, are current state of the part and have had quality control applied. However, as mentioned by Referee 1, the methods and information on errors are not given for all variables. I found it quite helpful to look at the LEFE-CYBER website*

*(http://www.obsvlfr.fr/proof/php/malina/x_datalist_1.php?xxop=malina&xxcamp=malin a) mentioned by the author as home for metadata and raw data, but see that they do not have information documents for all the variables. Looking through the data folders there are some documents that exist but have not been added as links to the site's variable list (ex. CTD documentation in basic files/doc and basic files/report), but in other cases appear missing, for example "Chlorophyll-a and Phaeopigments (concentration)". It sounds like this will be addressed as mentioned in author's answers A1 and A2 to Referee 1 and information will be added to table 1 as to where to find methods and error information.*

**Answer A5**

Thank you for the constructive comment. As the reviewer pointed out, this was mostly addressed in answers A1 and A2 by adding more information in table 1. For example, as evidenced by the reviewer, the phaeopigments do not have a link to

the metadata on the LEFE-CYBER website. However, there are now two new references in table 1 that provides all the details on the methods:

1. https://aslopubs.onlinelibrary.wiley.com/doi/full/10.1002/lno.10581
2. https://bg.copernicus.org/articles/10/5911/2013/

Along with the PI information, the reader can now contact the authors of these papers and request data access or additional methodological information. We also added information in table 1 whether the data is provided on CYBER/SEANOE or by contacting the PI's.

**Comment C6**

*Data availability: It would help to have Table 1., the heart of the paper with key information, expanded to state if the listed variables are present in the provided data set. For example, oxygen Winkler data are not present. I assume the list on the LEFE-CYBER site is current and shows the other missing data. Having a table of all the variables collected is very useful and a user can follow up by contacting the PI, however it would be good to know what is actually provided in the paper's data set.*

**Answer A6**

We agree that some data presented in Table 1 are not directly available in the dataset presented on SEANOE. This is why we have removed variables that are not available from Table 1. We also added information in Table 1 whether the data is directly available on SEANOE or by contacting the PI associated to the parameter (see the new column entitled *Included in the data repository*).

**Comment C7**

*Dataset: The large amount and various types of data pose a challenge in assembling a data collection. I did not find the current form of the final data posted to the SEANOE website useable due to its unwieldly size, structure and unclear variable names. A. The full data set is given in a single excel file approximately 2400 columns by 600,000 rows. The data set is too large to handle for software and users. My computer could open the file in excel but did not have enough memory to sort the rows. Its not practical for a user to sort through 2400 columns to find the variables of interest. A lookup table would help. This file could be separated into smaller files, in particular those data sets with a substantial number of columns (ANAP, AP and APHY with 500 columns each), and those that are not directly related (i.e. CTD, Foredeck meteorogical data, Benthic Boxcore, Zooplankton sampling). Remove other cruise data. There are data from programs in Aug 2008, Feb 2009, and the TARA programs from Jan 2010 to Aug 2012. Remove columns associated with these other programs (i.e. CTD data starting with column CNZ are for the TARA programs).*

**Answer A7**

As pointed out, the original file uploaded on SEANOE was a single large file containing all the merged data. We agree that it could be cumbersome to read with a spreadsheet software. Therefore, we are now providing a collection of smaller files that are regrouping all measurements associated with a type of measure. There is now approximately 50 comma separated values (CSV) files that contain the data. For example, there is now a single file containing only particulate absorption data. This will help future users to work with smaller datasets. Among these files, there is also a file called *stations.csv* that can serve as a lookup table to join the data altogether based on *date, time, station, cast, depth, longitude* and *latitude*. This

information has been added into the manuscript in section 6 (*Code and data availability*). We also double-checked that no other data than those of the MALINA cruise were included in the files.

**Comment C8**

*Variable names are not clear enough to determine what they are, which instrument, method or PI they are from, and what their unit of measurement is (i.e. "Average","345_1_AC"). Some variable names are shared between some data sets ("Cast", "Station", "Temperature", "Bottle"), some are not ("Temperature v. "Temp_Celsius", "Trans" v. "Xmiss", "Bottle" v. "N_btl_fired" v. "Niskin", "Station v. Q_Name"). In addition to clarifying what the variables and units are, it is important to be able to understand which variables connect to the information provided in the paper and Table 1. This currently is not practically possible. This could be addressed in a lookup table or with the variable name and header.*

**Answer A8**

We have reviewed all the data files to make sure that: (1) the file names were descriptive enough, so the reader can easily figure out which data is included and (2) that variable names inside the files (i.e. column names) were more descriptive. As stated in answer A7, we are now providing a lookup table (*stations.csv*) that can be used to merge data altogether.

**Comment C9**

*The many analyses from a given Niskin bottle do not use the same identifiers making it difficult to match all the data associated with a given water sample. The appended data set use a mix between Cast, Station, Bottle (and its variants), and Depth. At best these data would be pre-joined, but at least it would be good to have a consistent identifier.*

**Answer A9**

This issue has been resolved in answers A7 and A8.

**Comment A10**

*The date variable (column T) is a mixture of formats, only some currently understood by excel and thus not currently sortable. Corrections needed w/in data set: The CTD bottle data is one of the key parts to this data set that all water samples are linked back to. This data needs variable name "Depth_dbar" changed to "Pressure_dbar". The CTD+IOP data from the barge have been merged without any identifier to the date, cast, station or location. From the info sheet on the LEFE-CYBER page it looks like two steps are needed – the filename includes the IOP cast # and the infosheet has a table converting IOP cast # to station.*

**Answer A10**

Because the data has been divided into a set of smaller files, the column T (date) is no longer problematic. All data file has its own date column that should be recognized by any spreadsheet software programs or programming languages (R, Python, Matlab, etc.). Also, the date format provided in *stations.csv* compile to the ISO 8601 standard (ex:. 2009-08-03).

**Comment C11**

*Specific comments on article's content Sea-ice cover The paper mentions the sea ice in Line 49 and 50 (shelf was not ice-free until mid-August) and Line 99 (shelf was ice-free). These appear inconsistent and it would be good to harmonize these lines. Perhaps an image of sea-ice concentration could be added to the paper within the environmental conditions section or a sea-ice edge added to the map of stations. Station order Add a*

*further comment regarding the effect of sampling out of order. Did wind events occur between station sampling that may affect results? Even a statement saying this was not an issue or may be an issue would be helpful for a data user.*

**Answer A11**

We realize that details about sea ice information were missing. We have added a new figure (Fig. 2) that shows a MODIS true color image (Fig. 2A) and four weekly maps of sea ice concentration conditions (Fig. 2B) that prevailed during the MALINA oceanic cruise. Furthermore, we also revised the text accordingly to clarify what were the sea ice conditions during the cruise (for instance see the first paragraph in the section *2.2 General sampling strategy*).

**Comment C12**

*Technical Comments/Corrections Figure 8. Add "Surface samples" to the caption for clarity; It would be easier to compare if both sections were on the same plot.*

**Answer A12**

The caption has been changed accordingly. As for plotting the lines on the same graphs, the current presentation of the data comes from the PI responsible for the data. If the editor judge it so, we can review the figure.

**Comment C13**

*There are a number of "?" in the text that appear to be waiting for more information Figure 11a caption"?" , Line 270 "?", Line 279 "?", Line 282 "?"*

**Answer A13**

Thank you for pointing it out. These "?" symbols were present because of missing LaTeX bibliographical entries. This has been fixed.

**Comment C14**

*Here suggest using the pre-defined (Line 252) SCM instead of DCM for consistency.*

**Answer A14**

This has been corrected.

**Comment C15**

*Figure 14A Would be nice to see A and B use same latitude range.*

**Answer A15**

This has been done.

**Comment C16**

*Line 380 Looks like this is supposed to be Station 690, not 680*

**Answer A16**

The reviewer is right. This has been corrected.

**Comment C17**

*Line 489: Title is missing text. "A 50 % increase in the mass of terrestrial. . ."*

**Answer A17**

In our version of the PDF, it seems that the title of this article is correct:

*A 50 % increase in the mass of terrestrial particles delivered by the Mackenzie River into the Beaufort Sea (Canadian Arctic Ocean) over the last 10 years*

Please let us know if we misunderstood the comment.